# Polycomb repressive complex 1.1 coordinates homeostatic and emergency myelopoiesis

**Yaeko Nakajima-Takagi[1], Motohiko Oshima[1], Junichiro Takano[2], Shuhei Koide[1], Naoki Itokawa[1], Shun Uemura[1], Masayuki Yamashita[1], Shohei Andoh[1], Kazumasa Aoyama[1], Yusuke Isshiki[3], Daisuke Shinoda[1], Atsunori Saraya[3], Fumio Arai[4], Kiyoshi Yamaguchi[5], Yoichi Furukawa[5], Haruhiko Koseki[2], Tomokatsu Ikawa[6], Atsushi Iwama[1,7]***

[1]Division of Stem Cell and Molecular Medicine, Center for Stem Cell Biology and Regenerative Medicine, The Institute of Medical Science, University of Tokyo, Tokyo, Japan; [2]Laboratory for Developmental Genetics, RIKEN Center for Integrative Medical Sciences, Yokohama, Japan; [3]Department of Cellular and Molecular Medicine, Chiba University, Chiba, Japan; [4]Department of Stem Cell Biology and Medicine, Kyushu University, Fukuoka, Japan; [5]Division of Clinical Genome Research,Advanced Clinical Research Center, The Institute of Medical Science, University of Tokyo, Tokyo, Japan; [6]Division of Immunobiology, Research Institute for Biomedical Sciences, Tokyo University of Science, Chiba, Japan; [7]Laboratoty of Cellular and Molecular Chemistry, Graduate School of Pharmaceutical Sciences, University of Tokyo, Tokyo, Japan

**\*For correspondence:**
03aiwama@ims.u-tokyo.ac.jp

**Competing interest:** The authors declare that no competing interests exist.

**Abstract** Polycomb repressive complex (PRC) 1 regulates stem cell fate by mediating mono-ubiquitination of histone H2A at lysine 119. While canonical PRC1 is critical for hematopoietic stem and progenitor cell (HSPC) maintenance, the role of non-canonical PRC1 in hematopoiesis remains elusive. PRC1.1, a non-canonical PRC1, consists of PCGF1, RING1B, KDM2B, and BCOR. We recently showed that PRC1.1 insufficiency induced by the loss of PCGF1 or BCOR causes myeloid-biased hematopoiesis and promotes transformation of hematopoietic cells in mice. Here we show that PRC1.1 serves as an epigenetic switch that coordinates homeostatic and emergency hematopoiesis. PRC1.1 maintains balanced output of steady-state hematopoiesis by restricting C/EBPα-dependent precocious myeloid differentiation of HSPCs and the HOXA9- and β-catenin-driven self-renewing network in myeloid progenitors. Upon regeneration, PRC1.1 is transiently inhibited to facilitate formation of granulocyte-macrophage progenitor (GMP) clusters, thereby promoting emergency myelopoiesis. Moreover, constitutive inactivation of PRC1.1 results in unchecked expansion of GMPs and eventual transformation. Collectively, our results define PRC1.1 as a novel critical regulator of emergency myelopoiesis, dysregulation of which leads to myeloid transformation.

## Editor's evaluation

The authors present a manuscript aiming to understand the mechanism(s) underlying myeloid bias in HSCs, specifically focused on the role of Pcgf1, and therefore PRC1.1, in the regulation of hematopoiesis. The current study demonstrates that conditional deletion of Pcgf1 in HSCs causes a lineage switch from lymphoid to myeloid fates and that a key mechanism for this lineage switch is regulation of the H2AK119ub1 chromatin mark, leading to de-repression of CEBPalpha, a key transcription factor that promotes myeloid cell fate. This important work is of interest to the community

of researchers interested in myeloid differentiation, lineage fate decisions in hematopoietic stem cells, and the molecular mechanisms that contribute to the initiation of myeloid malignancies. The methods are rigorous and the results convincingly support the authors' conclusions.

## Introduction

While lifelong hematopoiesis is considered driven by hematopoietic stem cells (HSCs) (*Sawai et al., 2016*; *Säwen et al., 2018*), recent evidence pointed out a major role for multipotent progenitors (MPPs) and lineage-committed progenitors in hematopoiesis (*Busch et al., 2015*; *Sun et al., 2014*). Hematopoietic stem and progenitor cells (HSPCs) are highly responsive to various stresses such as infection, inflammation, and myeloablation (*Trumpp et al., 2010*; *Zhao and Baltimore, 2015*), which facilitate myelopoiesis by activating HSPCs to undergo precocious myeloid differentiation and transiently amplifying myeloid progenitors that rapidly differentiate into mature myeloid cells (*Hérault et al., 2017*; *Pietras et al., 2016*). This reprogramming of HSPCs, termed 'emergency myelopoiesis,' serves to immediately replenish mature myeloid cells to control infection and regeneration (*Manz and Boettcher, 2014*). Recent evidence further suggested that uncontrolled activation of the myeloid regeneration programs results in the development of chronic inflammatory diseases and hematological malignancies (*Chiba et al., 2018*; *Zhao and Baltimore, 2015*). Emergency myelopoiesis is driven via activation of key myeloid transcriptional networks at the HSPC and myeloid progenitor cell levels (*Rosenbauer and Tenen, 2007*). However, the epigenetic regulatory mechanisms governing emergency myelopoiesis remained largely unknown.

Polycomb group (PcG) proteins are the key epigenetic regulators of a variety of biological processes (*Piunti and Shilatifard, 2021*). They comprise the multiprotein complexes, polycomb repressive complex (PRC) 1 and PRC2, which establish and maintain the transcriptional repression through histone modifications. PRC1 and PRC2 add mono-ubiquitination at lysine 119 of histone H2A (H2AK119ub) and mono-, di-, and tri-methylation at lysine 27 of histone H3 (H3K27me1/me2/me3), respectively, and cooperatively repress transcription (*Blackledge et al., 2015*; *Iwama, 2017*). PRC1 complexes are divided into subgroups (PRC1.1 to PRC1.6) according to the subtype of the Polycomb group ring finger (PCGF) subunits (PCGF1-6). PCGF2/MEL18 and PCGF4/BMI1 act as components of canonical PRC1 (PRC1.2 and 1.4, respectively) that are recruited to its target sites in a manner dependent on H3K27me3, whereas the others (PCGF1, 3, 5, and 6) constitute non-canonical PRC1 (PRC1.1, PRC1.3, PRC1.5, and PRC1.6, respectively) that are recruited independently of H3K27me3 (*Blackledge et al., 2014*; *Gao et al., 2012*; *Wang et al., 2004*).

PCGF4/BMI1-containing canonical PRC1 (PRC1.4) has been characterized for its role in maintaining self-renewal capacity and multipotency of HSCs (*Sashida and Iwama, 2012*). We and others have reported that BMI1 transcriptionally represses the loci for *CDKN2A* and developmental regulator genes (e.g., B cell regulators) to maintain self-renewal capacity and multipotency of HSPCs (*Iwama et al., 2004*; *Oguro et al., 2010*; *Park et al., 2003*). We also reported that PCGF5-containing PRC1.5 regulates global levels of H2AK119ub, but is dispensable for HSPC function (*Si et al., 2016*). On the other hand, we and others recently showed that PRC1.1 components, PCGF1, KDM2B, and BCOR, maintain normal hematopoiesis and suppress malignant transformation of hematopoietic cells (*Andricovich et al., 2016*; *Isshiki et al., 2019*; *Tara et al., 2018*). PRC1.1 consists of PCGF1, RING1A/B, KDM2B, and BCOR or BCLRL1. KDM2B binds to non-methylated CpG islands through its DNA-binding domain, thereby recruiting other PRC1.1 components (*Farcas et al., 2012*; *He et al., 2013*). PCGF1 was found to restrict the proliferative capacity of myeloid progenitor cells by downregulating *Hoxa* family genes in in vitro knockdown experiments (*Ross et al., 2012*). Correspondingly, we demonstrated that Pcgf1 loss induces myeloid-biased hematopoiesis and promotes JAK2V617F-induced myelofibrosis in mice (*Shinoda et al., 2022*). Bcor loss also showed myeloid-biased hematopoiesis and promoted the initiation and progression of myelodysplastic syndrome in collaboration with Tet2 loss (*Tara et al., 2018*). However, detailed analysis of the role for PRC1.1 in hematopoiesis, especially in the context of hematopoietic regeneration and emergency myelopoiesis, has not yet been reported.

Here, we analyzed the murine hematopoiesis in the absence of PCGF1 and uncovered critical roles of PCGF1-containing PRC1.1 in homeostatic, emergency, and malignant hematopoiesis.

# Results

## PCGF1 restricts myeloid commitment of HSPCs

To understand the function of PCGF1 in hematopoiesis, we crossed *Pcgf1^fl* mice, in which exons 2–7 are floxed (*Figure 1—figure supplement 1A*), with *Rosa26::Cre-ERT2* (*Rosa26^CreERT*) mice (*Rosa-26^CreERT;Pcgf1^fl/fl*). To delete *Pcgf1* specifically in hematopoietic cells, we transplanted bone marrow (BM) cells from *Rosa26^CreERT* control and *Rosa26^CreERT;Pcgf1^fl/fl* CD45.2 mice into lethally irradiated CD45.1 recipient mice and deleted *Pcgf1* by intraperitoneal injection of tamoxifen (*Figure 1A* and *Figure 1—figure supplement 1B*). We confirmed the efficient deletion of *Pcgf1* in donor-derived hematopoietic cells from the peripheral blood (PB) by genomic PCR (*Figure 1—figure supplement 1C*). RT-qPCR confirmed the significant reduction of *Pcgf1* mRNA lacking exons 2–7 in donor-derived BM Lineage marker⁻Sca-1⁺c-Kit⁺ (LSK) HSPCs (*Figure 1—figure supplement 1D*). We hereafter refer to the recipient mice reconstituted with *Rosa26^CreERT* control and *Rosa26^CreERT;Pcgf1^fl/fl* cells treated with tamoxifen as control and *Pcgf1^Δ/Δ* mice, respectively.

*Pcgf1^Δ/Δ* mice exhibited mild anemia and leukopenia, which was mainly attributed to the reduction in B cell numbers (*Figure 1B*, *Figure 1—figure supplement 2A and B*). Myeloid cell numbers in PB were relatively maintained and their proportion increased over time in *Pcgf1^Δ/Δ* mice (*Figure 1B* and *Figure 1—figure supplement 2B*). While BM cellularity was comparable between control and *Pcgf1^Δ/Δ* mice, Mac1⁺ mature myeloid cells were increased at the expense of B cells in *Pcgf1^Δ/Δ* BM and spleen (*Figure 1C* and *Figure 1—figure supplement 2C–E*). Monocytes but not neutrophils or F4/80⁺ macrophages significantly increased in *Pcgf1^Δ/Δ* BM while both neutrophils and monocytes significantly increased in *Pcgf1^Δ/Δ* spleen (*Figure 1—figure supplement 2C and D*). These findings indicate that *Pcgf1^Δ/Δ* HSPCs generate more mature myeloid cells than the control without showing evident differentiation block. Among the committed progenitor cells, the numbers of granulocyte-macrophage progenitors (GMPs) were significantly increased in *Pcgf1^Δ/Δ* BM, whereas those of megakaryocyte-erythroid progenitors (MEPs), pre- and pro-B cells, all-lymphoid progenitors (ALPs) and B-cell-biased lymphoid progenitors (BLPs) were decreased (*Figure 1C*, *Figure 1—figure supplement 2C*, and *Figure 1—figure supplement 3B*). Of interest, *Pcgf1^Δ/Δ* mice showed reduction in the numbers of HSCs and all subsets of MPPs in BM (*Figure 1C* and *Figure 1—figure supplement 3A*). The reduction in the HSPC pool size was accompanied by decreased cells in cycling phases in *Pcgf1^Δ/Δ* HSPCs (CD34⁻ and CD34⁺ LSK cells) (*Figure 1D*). Despite the reduction in phenotypic HSPCs, limiting dilution assays with competitive BM transplantation revealed that the numbers of HSPCs that established long-term repopulation of myeloid and T cells (B cells were excluded due to the low contribution of *Pcgf1^Δ/Δ* HSPCs to B cells, see below) were comparable between control and *Pcgf1^Δ/Δ* BM (*Figure 1E*). Extramedullary hematopoiesis was evident in the *Pcgf1^Δ/Δ* spleen, as judged by markedly increased numbers of LSK HSPCs, GMPs, and mature myeloid cells including monocytes and neutrophils (*Figure 1—figure supplement 2C, D, and F* and *Figure 1—figure supplement 3C*). Differentiation of thymocytes in the *Pcgf1^Δ/Δ* thymus was largely normal (*Figure 1—figure supplement 2C*).

To further evaluate the role of PCGF1 in hematopoiesis, we transplanted BM cells with the same number of CD45.1 wild-type (WT) competitor cells (*Figure 1F*). In this competitive setting, only a mild decrease was detected in the overall chimerism of CD45.2⁺ *Pcgf1^Δ/Δ* cells in PB (*Figure 1G*). In contrast, the chimerism of *Pcgf1^Δ/Δ* cells in myeloid cells (Mac-1⁺ and/or Gr1⁺) was markedly increased while that in B cell lineage (B220⁺) was decreased (*Figure 1G*). In BM, *Pcgf1^Δ/Δ* cells outcompeted the competitor cells in the myeloid lineage compartments from the common myeloid progenitor (CMP) stage (*Figure 1H*). Since *Pcgf1^Δ/Δ* cell showed reductions in the numbers of HSCs and MPPs in a non-competitive setting (*Figure 1C*), we examined the absolute numbers of test and competitor cells in this competitive repopulation. Of interest, the competitive *Pcgf1^Δ/Δ* recipients also exhibit similar changes in BM hematopoietic cell numbers. Both CD45.2⁺ *Pcgf1^Δ/Δ* and CD45.1⁺ WT cells were depleted in HSPC fractions in *Pcgf1^Δ/Δ* recipients, while the total numbers of myeloid progenitors and mature myeloid cells were maintained or rather increased (*Figure 1I*). These findings suggest that *Pcgf1^Δ/Δ* hematopoietic cells suppress expansion of both *Pcgf1^Δ/Δ* and co-existing WT HSPCs through non-autonomous mechanisms. It is possible that accumulating *Pcgf1^Δ/Δ* mature myeloid cells produce inflammatory cytokines that are suppressive to HSPCs. The underlying mechanism is an intriguing issue to be analyzed. To evaluate the impact of PCGF1 loss on long-term hematopoiesis, we harvested BM cells from primary recipient mice 4 mo after tamoxifen injection and transplanted them

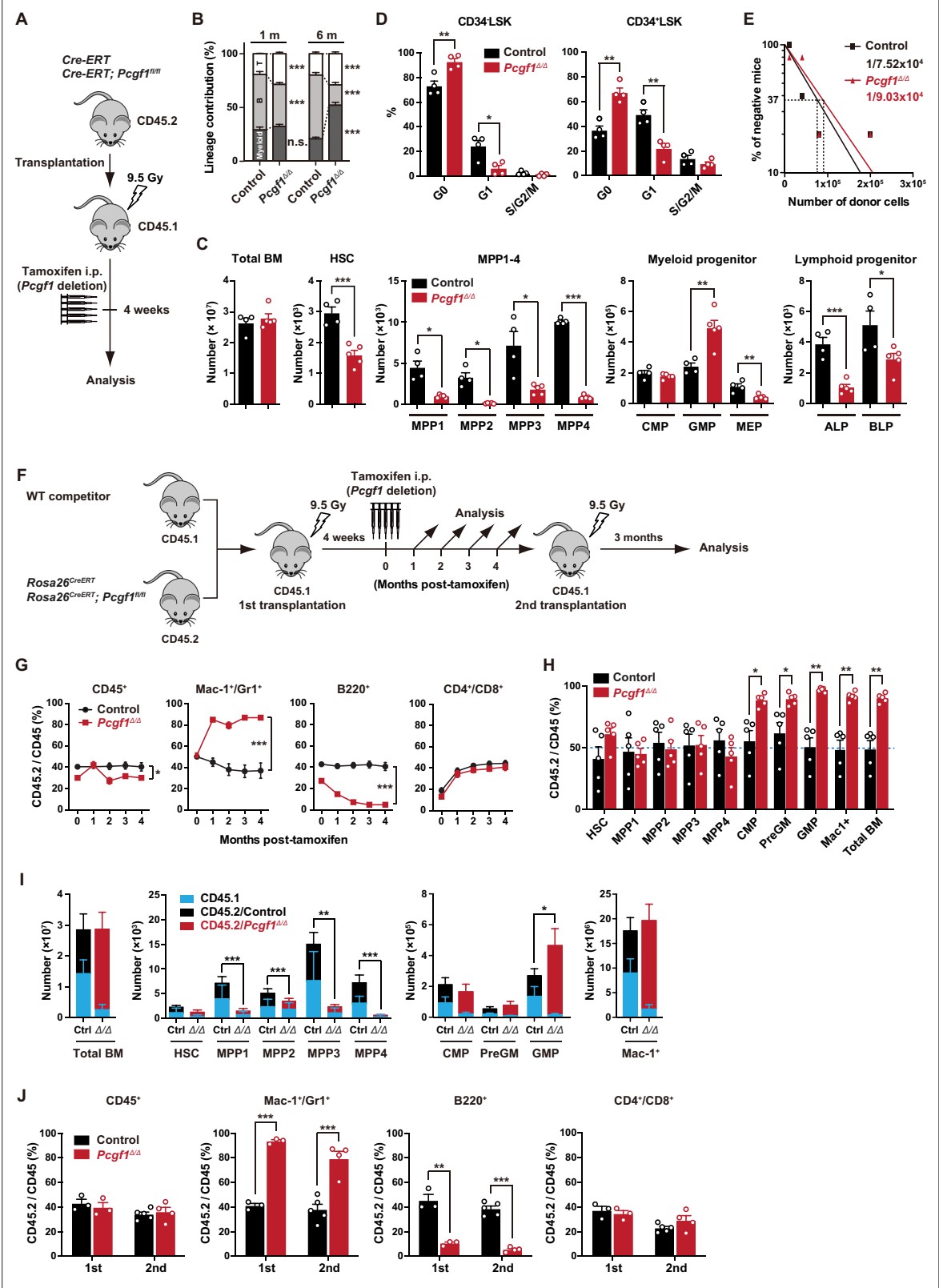

**Figure 1.** PCGF1 regulates myelopoiesis but not self-renewal of hematopoietic stem and progenitor cells (HSPCs). (**A**) Strategy for analyzing *Pcgf1^Δ/Δ* hematopoietic cells. Total bone marrow (BM) cells (5 × 10⁶) from *Rosa26^CreERT* and *Rosa26^CreERT*;*Pcgf1^fl/fl* were transplanted into lethally irradiated CD45.1 recipient mice. *Pcgf1* was deleted by intraperitoneal injections of tamoxifen at 4 wk post-transplantation. (**B**) The proportions of Mac-1⁺ and/or Gr-1⁺ myeloid cells, B220⁺ B cells, and CD4⁺ or CD8⁺ T cells among CD45.2⁺ donor-derived hematopoietic cells in the peripheral blood (PB) from control

*Figure 1 continued on next page*

*Figure 1 continued*

(n = 9) and *Pcgf1^Δ/Δ* (n = 14) mice. (**C**) Absolute numbers of total BM cells, hematopoietic stem cells (HSCs), multipotent progenitors (MPPs), myeloid progenitors, and CLPs (ALP and BLP) in a unilateral pair of femur and tibia 4 wk after the tamoxifen injection (n = 4–5). (**D**) Cell cycle status of CD34⁻LSK HSCs and CD34⁺LSK MPPs assessed by Ki67 and 7-AAD staining 4 wk after the tamoxifen injection. (**E**) In vivo limiting dilution assay. Limiting numbers of BM cells (1 × 10⁴, 4 × 10⁴, 8 × 10⁴, and 2 × 10⁵) isolated from BM of primary recipients (control and *Pcgf1^Δ/Δ* mice after transplantation) were transplanted into sublethally irradiated secondary recipient mice with 2 × 10⁵ of competitor CD45.1 BM cells (n = 5 each). Due to the low contribution of *Pcgf1^Δ/Δ* HSPCs to B cells, mice with chimerism of donor myeloid and T cells more than 1% in the PB at 16 wk after transplantation were considered to be engrafted successfully, and the others were defined as non-engrafted mice. The frequencies of HSPCs that contributed to both myeloid and T cells are indicated. (**F**) Strategy for analyzing *Pcgf1^Δ/Δ* hematopoietic cells. Total BM cells (2 × 10⁶) from *Rosa26^CreERT* and *Rosa26^CreERT;Pcgf1^fl/fl* CD45.2 mice were transplanted into lethally irradiated CD45.1 recipient mice with the same number of competitor CD45.1 BM cells. *Pcgf1* was deleted by intraperitoneal injections of tamoxifen at 4 wk post-transplantation. Secondary transplantation was performed using 5 × 10⁶ total BM cells from primary recipients at 4 mo post-intraperitoneal injections of tamoxifen. (**G**) The chimerism of CD45.2 donor cells in PB CD45⁺ leukocytes, Mac-1⁺ and/or Gr1⁺ myeloid cells, B220⁺ B cells, and CD4⁺ or CD8⁺ T cells in control and *Pcgf1^Δ/Δ* mice (n = 6 each) after the tamoxifen injection. (**H**) The chimerism of CD45.2 donor-derived cells in BM 4 wk after the tamoxifen injection (n = 5). (**I**) Absolute numbers of CD45.1 and CD45.2 total BM cells, HSCs, MPPs, myeloid progenitors, and Mac-1⁺ mature myeloid cells in a unilateral pair of femur and tibia 4 wk after the tamoxifen injection (n = 5). Statistical significance is based on the overall number of cells. (**J**) The chimerism of CD45.2 donor-derived cells in PB in primary (n = 3 each) and secondary (n = 4–5) transplantation. Data are shown as the mean ± SEM. *p<0.05, **p<0.01, ***p<0.001 by the Student's *t*-test. Each symbol is derived from an individual mouse. A representative of more than two independent experiments is shown.

The online version of this article includes the following source data and figure supplement(s) for figure 1:

**Source data 1.** Raw data for *Figure 1*.

**Figure supplement 1.** Targeting of the *Pcgf1* gene in the mouse hematopoietic system.

**Figure supplement 1—source data 1.** Uncropped gel images of *Figure 1—figure supplement 1C*.

**Figure supplement 1—source data 2.** Raw data for *Figure 1—figure supplement 1*.

**Figure supplement 2.** Accumulation of myeloid cells in *Pcgf1^Δ/Δ* mice.

**Figure supplement 2—source data 1.** Raw data for *Figure 1—figure supplement 2*.

**Figure supplement 3.** Flow cytometric profiles of *Pcgf1^Δ/Δ* hematopoietic stem and progenitor cells (HSPCs) and lymphoid progenitors.

**Figure supplement 4.** Growth and differentiation of *Pcgf1^Δ/Δ* hematopoietic stem cells (HSCs) in culture.

**Figure supplement 4—source data 1.** Raw data for *Figure 1—figure supplement 4*.

---

into secondary recipients. *Pcgf1^Δ/Δ* cells reproduced the myeloid-biased hematopoiesis in secondary recipients (*Figure 1J*).

We next evaluated the capacity of *Pcgf1^Δ/Δ* HSCs to differentiate to myeloid and lymphoid cells in culture. *Pcgf1^Δ/Δ* HSCs displayed slower population doubling under HSPC-expanding culture conditions (*Figure 1—figure supplement 4A*), which is in good agreement with fewer cycling *Pcgf1^Δ/Δ* HSPCs (*Figure 1D*). Nevertheless, *Pcgf1^Δ/Δ* HSCs showed better population doubling than control cells under myeloid culture conditions (*Figure 1—figure supplement 4A*). On the other hand, limiting dilution assays using a co-culture system with TSt-4 stromal cells (*Masuda et al., 2005*) revealed that the capacity of *Pcgf1^Δ/Δ* HSCs to produce B and T cells was declined by two- and fivefold, respectively, compared to the control (*Figure 1—figure supplement 4B*). The discrepancy in T cell production between in vitro and in vivo may be due to the compensatory expansion of T cells in the thymus. These results indicate that PCGF1 loss enhances myelopoiesis at the expense of lymphopoiesis. These phenotypes were similar to those of mice expressing a carboxyl-terminal truncated BCOR that cannot interact with PCGF1 (*Tara et al., 2018*).

## PCGF1 inhibits precocious myeloid commitment of HSPCs through repression of C/EBPα, which is critical for balanced output of HSPCs

To clarify the molecular mechanisms underlying myeloid-biased differentiation of *Pcgf1^Δ/Δ* HSPCs, we performed RNA sequence (RNA-seq) analysis of HSPCs from mice 4 wk after the tamoxifen injection. Principal component analysis (PCA) showed shifts of the transcriptomic profiles of *Pcgf1^Δ/Δ* MPP2 and MPP3 toward pre-GMs (*Figure 2A*). Gene set enrichment analysis (GSEA) revealed upregulation of genes involved in myeloid cell development (*Brown et al., 2006*), CEBP targets (*Gery et al., 2005*), and genes upregulated upon C/EBPα overexpression (*Loke et al., 2018*) in *Pcgf1^Δ/Δ* HSPCs compared to controls (*Figure 2B* and *Supplementary file 1*). C/EBP family are master transcription factors for myeloid differentiation (*Avellino and Delwel, 2017*; *Rosenbauer and Tenen, 2007*). RT-PCR analysis

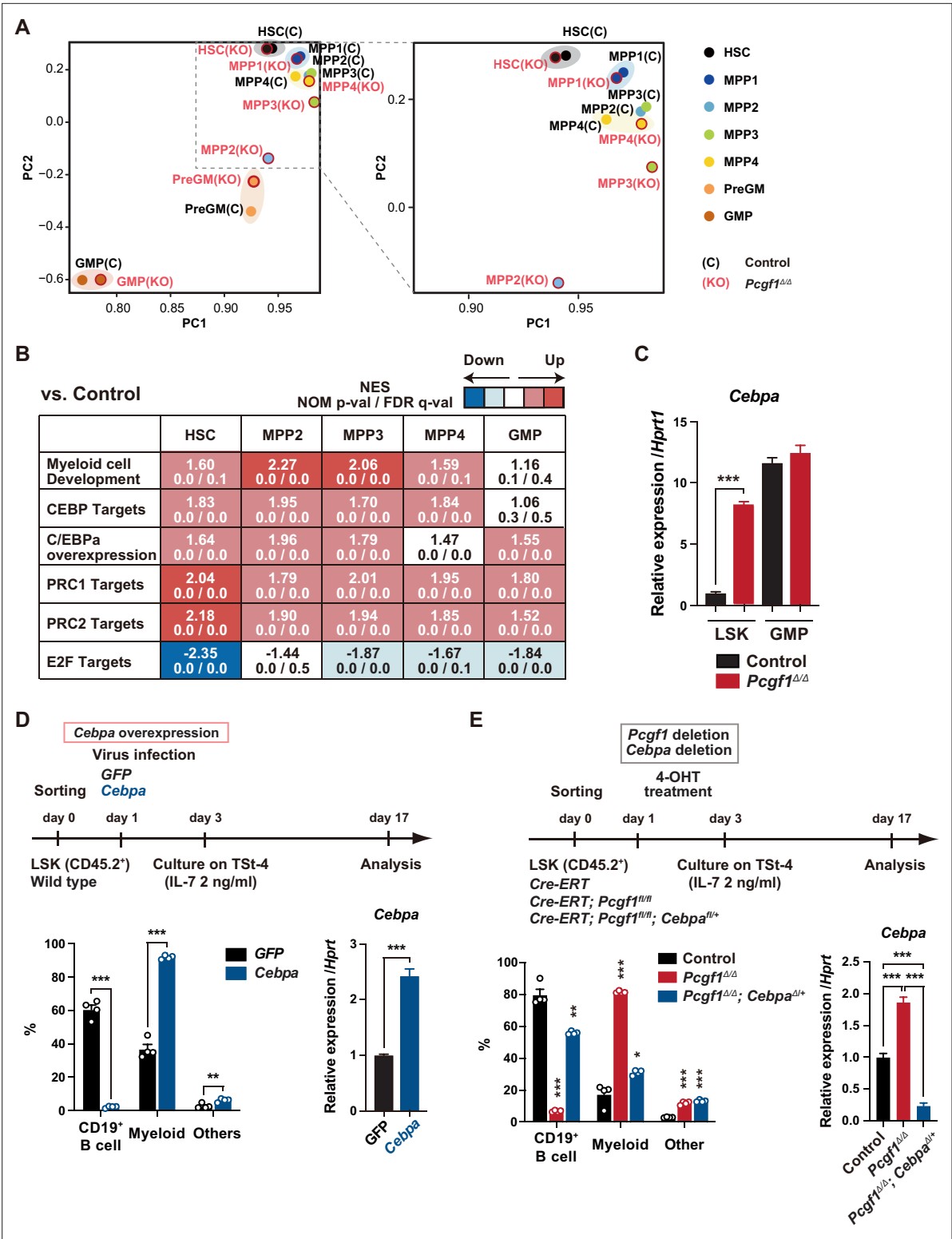

**Figure 2.** *Pcgf1*-deficient hematopoietic stem and progenitor cells (HSPCs) undergo myeloid reprogramming. (**A**) Principal component analyses (PCA) based on total gene expression obtained by RNA-seq of hematopoietic stem cells (HSCs), multipotent progenitors (MPPs), pre-GM, and granulocyte-macrophage progenitors (GMPs) from control and *Pcgf1*ᐞ/ᐞ mice. Magnified view of the boxed part is depicted on the right. (**B**) Gene set enrichment analysis (GSEA) using RNA-seq data. Summary of GSEA data of representative gene sets is shown. Normalized enrichment scores (NES), nominal p-values (NOM), and false discovery rates (FDR) are indicated. The gene sets used are indicated in ***Supplementary file 1***. (**C**) Quantitative RT-PCR analysis of *Cebpa* in LSK cells and GMPs. *Hprt1* was used to normalize the amount of input RNA. Data are shown as the mean ± SEM (n = 3). ***p<0.001 by

*Figure 2 continued on next page*

*Figure 2 continued*

the Student's *t*-test. (**D**) Effects of *Cebpa* overexpression on HSPC differentiation in vitro. LSK cells were transduced with either control (*GFP*) or *Cebpa* retrovirus, then cultured on TSt-4 stromal cells in the presence of IL-7 (upper). The proportions of myeloid (Mac1$^+$ and /or Gr-1$^+$), B cells (CD19$^+$), and others (Mac1$^-$Gr-1$^-$CD19$^-$) among CD45.2$^+$GFP$^+$ hematopoietic cells on day 17 of culture are indicated (lower left; n = 4 each). RT-qPCR analysis of *Cebpa* in LSK cells transduced with control or *Cebpa* retrovirus on day 14 of culture (n = 3). *Hprt1* was used to normalize the amount of input RNA (lower right). Each symbol is derived from an individual culture. Data are shown as the mean ± SEM. **p<0.01, ***p<0.001 by the Student's *t*-test. (**E**) Impact of *Cebpa* haploinsufficiency on myeloid-biased differentiation of *Pcgf1$^{Δ/Δ}$* HSPCs. LSK cells from *Rosa26$^{CreERT}$*, *Rosa26$^{CreERT}$;Pcgf1$^{fl/fl}$* and *Rosa26$^{CreERT}$;Pcgf1$^{fl/fl}$;Cebpa$^{fl/+}$* mice were treated with 4-OHT (200 nM) for 2 d in culture to delete *Pcgf1* and *Cebpa*. Cells were further cultured on TSt-4 stromal cells in the presence of IL-7 (upper). The proportions of myeloid (Mac1$^+$ and /or Gr-1$^+$), B cells (CD19$^+$), and others (Mac1$^-$Gr-1$^-$CD19$^-$) among CD45.2$^+$GFP$^+$ hematopoietic cells on day 17 of culture are indicated (lower; n = 4 each ; * versus control). Each symbol is derived from an individual culture (lower left). RT-qPCR data of *Cebpa* in LSK cells on day 14 of culture (n = 3). *Hprt1* was used to normalize the amount of input RNA (lower right). Data are shown as the mean ± SEM. *p<0.05, **p<0.01, ***p<0.001 by the Student's *t*-test (lower left) or the one-way ANOVA (lower right).

The online version of this article includes the following source data and figure supplement(s) for figure 2:

**Source data 1.** Raw data for *Figure 2*.

**Figure supplement 1.** Derepressed *Cebpa* accounts for the myeloid-biased differentiation of *Pcgf1$^{Δ/Δ}$* hematopoietic stem and progenitor cells (HSPCs).

**Figure supplement 1—source data 1.** Uncropped gel images of *Figure 2—figure supplement 1A*.

**Figure supplement 1—source data 2.** Raw data for *Figure 2—figure supplement 1*.

demonstrated a significant upregulation of *Cepba* in HSPCs, but not in GMPs (*Figure 2C*). C/EBPα drives myelopoiesis and antagonizes lymphoid differentiation (*Fukuchi et al., 2006*; *Rosenbauer and Tenen, 2007*; *Xie et al., 2004*). Disruption of *Cebpa* blocks the transition from CMPs to GMPs (*Zhang et al., 1997*). So, we evaluated the contribution of de-repressed *Cebpa* in *Pcgf1$^{Δ/Δ}$* HSPCs. First, we overexpressed *Cebpa* in WT LSK cells and cultured them on TSt-4 stromal cells. Only twofold upregulation of *Cebpa* was sufficient to enhance myeloid differentiation and suppress B cell differentiation of HSPCs (*Figure 2D*). Conversely, we next tested whether myeloid skewing of *Pcgf1$^{Δ/Δ}$* HSPCs could be canceled by reducing *Cebpa* expression. We seeded HSPCs from *Rosa26$^{CreERT}$*, *Rosa26$^{CreERT}$;Pcgf1$^{fl/fl}$*, and *Rosa26$^{CreERT}$;Pcgf1$^{fl/fl}$Cebpa$^{fl/+}$* mice on TSt-4 stromal cells and recombined *Pcgf1$^{fl}$* and *Cebpa$^{fl}$* alleles by supplementation of 4-hydroxytamoxifen (4-OHT) (*Figure 2—figure supplement 1A*). Of note, the reduction in the levels of *Cebpa* expression by introducing a *Cebpa$^{fl/+}$* allele in *Pcgf1$^{Δ/Δ}$* LSK cells was sufficient to restore balanced production of myeloid and B cells by *Pcgf1$^{Δ/Δ}$* HSPCs (*Figure 2E*). These results demonstrate that de-repression of *Cebpa* largely accounts for at least the myeloid-biased differentiation of *Pcgf1$^{Δ/Δ}$* HSPCs.

We next attempted to rescue the myeloid-biased differentiation of *Pcgf1$^{Δ/Δ}$* HSPCs by exogenous *Pcgf1* or a canonical PRC1 gene *Bmi1/Pcgf4*. We transduced *Rosa26$^{CreERT}$;Pcgf1$^{fl/fl}$* HSPCs to induce *Pcgf1* or *Bmi1* expression, transplanted them into lethally irradiated mice, and deleted endogenous *Pcgf1* (*Figure 2—figure supplement 1B*). The myeloid skew in *Pcgf1$^{Δ/Δ}$* PB leukocytes was completely prevented by ectopic expression of *Pcgf1* but not of *Bmi1* (*Figure 2—figure supplement 1B*), highlighting distinct roles of non-canonical PRC1.1 and canonical PRC1 in hematopoietic differentiation.

We also noticed that the E2F targets (*Ishida et al., 2001*) were downregulated in *Pcgf1$^{Δ/Δ}$* HSPCs (*Figure 2B*), which may underlie the disturbed cell cycle progression and delayed proliferation observed in *Pcgf1$^{Δ/Δ}$* HSPCs (*Figure 1D* and *Figure 1—figure supplement 4A*). C/EBPα represses E2F-mediated transcription (*D'Alo' et al., 2003*; *Slomiany et al., 2000*), inhibits HSC cell cycle entry (*Ye et al., 2013*; *Zhang et al., 2004*), and promotes precocious IL-1β-driven emergency myelopoiesis (*Higa et al., 2021*). Thus, upregulated *Cebpa* upon PCGF1 loss may inhibit cell cycle and promote myeloid commitment of HSPCs.

## Deletion of *Pcgf1* affects levels of H2AK119ub1

To understand how PCGF1 loss affects H2AK119ub1 status in HSPCs, we performed chromatin immunoprecipitation followed by sequencing (ChIP-seq) analysis using control and *Pcgf1$^{Δ/Δ}$* HSPCs. Since none of the anti-PCGF1 antibodies were suitable for ChIP analysis, we used 3×Flag-PCGF1-expressing BM LK cells obtained by retrovirally transducing LSK cells and transplanting them to lethally irradiated mice. We defined 'PRC1 targets' and 'PRC2 targets' as genes with H2AK119ub1 and H3K27me3 enrichment greater than twofold over the input signals in control LSK cells at promoter regions (transcription start site [TSS] ± 2.0 kb), respectively (*Supplementary file 2*). GSEA revealed that both PRC1

and PRC2 targets were upregulated in *Pcgf1*$^{\Delta/\Delta}$ HSPCs and GMPs (*Figure 2B*). K-means clustering divided PRC1 targets into two clusters depending on the levels of H2AK119ub1 and H3K27me3. Cluster 1 genes (1835 RefSeq ID genes) were marked with high levels of H2AK119ub1 and H3K27me3 at promoter regions, while cluster 2 genes (2691 RefSeq ID genes) showed moderate levels of H2AK119ub1 and H3K27me3 (*Figure 3A*). PCGF1 showed stronger binding to the promoters of cluster 2 genes than cluster 1 genes (*Figure 3A*). Interestingly, only cluster 2 genes showed moderate but significant reductions in H2AK119ub1 and H3K27me3 levels in *Pcgf1*$^{\Delta/\Delta}$ HSPCs (*Figure 3B*). The loss of PCGF1 was also significantly associated with de-repression of PRC1 target genes in clusters 1 and 2 (*Figure 3B*). These results suggest that cluster 2 genes are the major targets of PCGF1-containing PRC1.1 and *Cebpa* was included in cluster 2 genes. Bivalent genes, which are enriched for developmental regulator genes marked with both active and repressive histone marks (H3K4me3 and H3K27me3, respectively, mostly with H2AK119ub1), are classical targets of canonical PRC1 and PRC2 (*Bernstein et al., 2006*; *Ku et al., 2008*) and are implicated in multipotency of HSPCs (*Oguro et al., 2010*). Of note, bivalent genes defined by our previous ChIP-seq data of HSPCs (*Aoyama et al., 2018*; *Supplementary file 2*) were more enriched in cluster 1 genes than cluster 2 genes (*Figure 3C*).

We then defined 'PCGF1 targets' whose H2AK119ub1 levels were decreased by *Pcgf1* deletion greater than twofold at promoter regions in HSPCs (997 RefSeq ID genes; *Supplementary file 2*). We found that *Cebpa and Cebpe* were included in PCGF1 targets, showed reductions in H2AK119ub1 levels (*Figure 3D and E*), and were de-repressed in expression in *Pcgf1*$^{\Delta/\Delta}$ LSK cells (*Figure 2C* and *Figure 2—figure supplement 1C*). ChIP-qPCR confirmed a significant reduction in H2AK119ub1 levels at the promoter region of *Cebpa* in *Pcgf1*$^{\Delta/\Delta}$ HSPCs (*Figure 3F*). These results indicate that the deletion of *Pcgf1* compromises PRC1.1 function and causes precocious activation of key myeloid regulator genes in HSPCs.

To clarify whether the *Pcgf1* deletion has any impact on the chromatin accessibility in HSPCs, we performed an assay for transposase accessible chromatin with high-throughput sequencing (ATAC-seq) analysis using CD135⁻LSK HSPCs, which include HSCs and MPP1-3 but not lymphoid-primed MPP4. ATAC-seq profiles open chromatin regions enriched for transcriptional regulatory regions, such as promoters and enhancers. ATAC peaks were significantly enriched at the promoter regions of PCGF1 target genes, but not of PCGF1 non-target genes, in *Pcgf1*$^{\Delta/\Delta}$ HSPCs compared to the corresponding controls (*Figure 3G*), further validating de-repression of PCGF1 targets upon the deletion of *Pcgf1*. Motif analysis of ATAC peaks in the proximal promoter regions (TSS ± 2 kb) revealed that the CTCF motif, which has been reported to be associated with differentiation of HSCs (*Buenrostro et al., 2018*; *Takayama et al., 2021*), was highly enriched in *Pcgf1*$^{\Delta/\Delta}$ HSPCs (*Figure 3H*). Interestingly, the other top-ranked motifs were related to stress response transcription factors, such as Bach family (Bach1 and 2), CNC family (Nrf2 and NF-E2), AP1 family, and Atf3 (*Figure 3H* and *Supplementary file 3*). In contrast, the binding motifs for transcription factors essential for T and B cell commitment, including HEB and E2A (*de Pooter and Kee, 2010*), were negatively enriched in *Pcgf1*$^{\Delta/\Delta}$ HSPCs (*Figure 3H*). Together with the significant upregulation of *Cebpa* and *Cebpe* in *Pcgf1*$^{\Delta/\Delta}$HSPCs (*Figure 2C* and *Figure 2—figure supplement 1C*), these results further support the notion that PRC1.1 deficiency caused precocious activation of myeloid differentiation program at the expense of lymphoid differentiation program in HSPCs.

## PCGF1 inhibition facilitates emergency myelopoiesis

Myeloid-biased hematopoiesis in *Pcgf1*-deficient mice reminded us of the myeloproliferative reactions caused by emergencies such as regeneration (*Manz and Boettcher, 2014*). Individual GMPs scatter throughout the BM in the steady state, while expanding GMPs evolve into GMP clusters during regeneration, which, in turn, differentiate into granulocytes. Inducible activation of β-catenin and *Irf8* controls the formation and differentiation of GMP clusters, respectively (*Hérault et al., 2017*). Immunofluorescence analyses of BM sections readily detected GMP clusters in steady-state *Pcgf1*$^{\Delta/\Delta}$ BM (*Figure 4A*), which was reminiscent of those observed during regeneration after 5-fluorouracil (5-FU) treatment (*Figure 4—figure supplement 1A*). To address whether PCGF1 also regulates myelopoiesis at the GMP level during regeneration, we challenged control and *Pcgf1*$^{\Delta/\Delta}$ mice with a single dose of 5-FU (*Figure 4B*). Consistent with the previous report (*Hérault et al., 2017*), control mice showed transient expansion of BM HSPCs and GMPs around day 14 and subsequent burst of circulating PB myeloid cells around day 21. In sharp contrast, *Pcgf1*$^{\Delta/\Delta}$ mice displayed sustained GMP expansion until

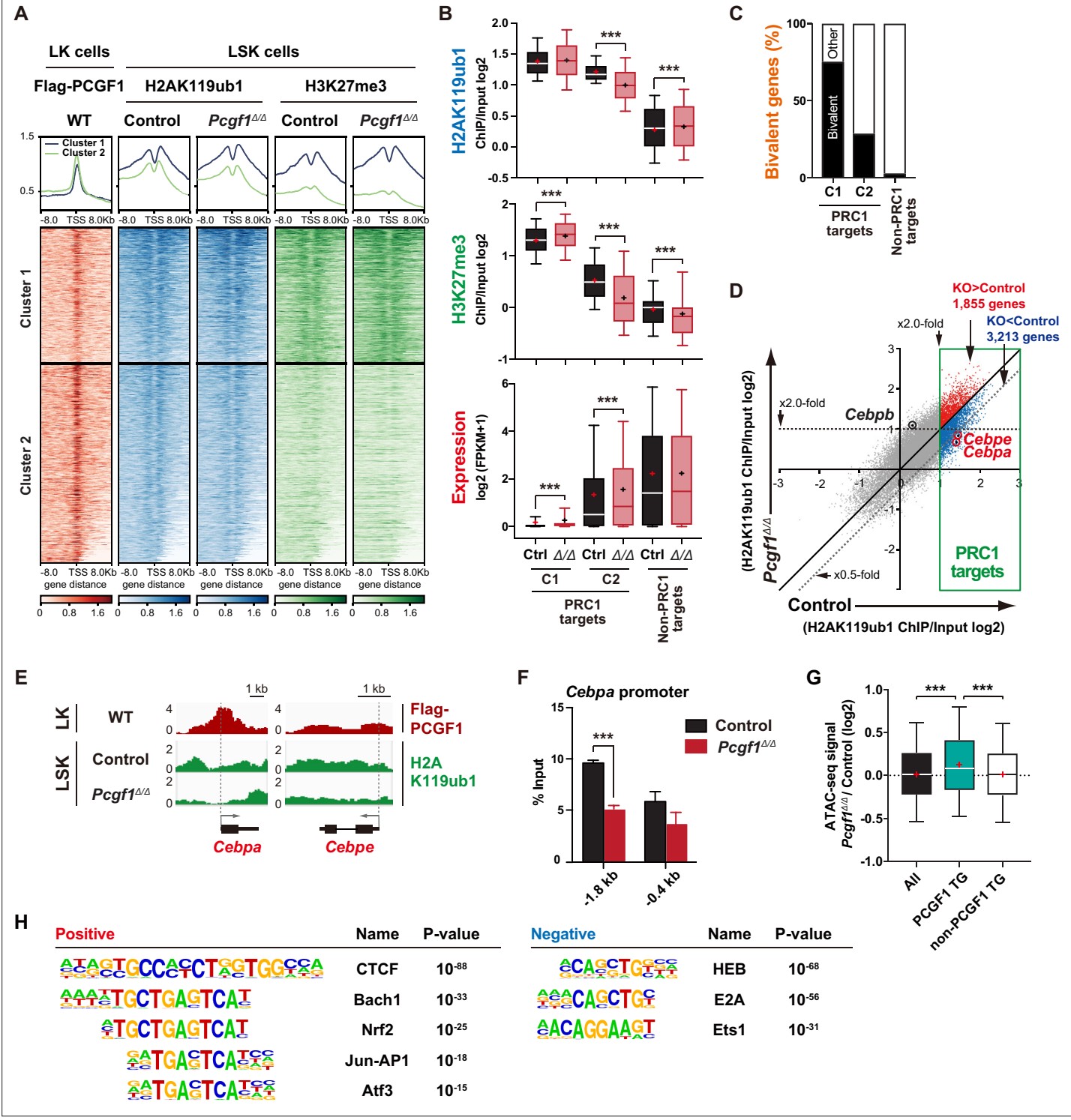

**Figure 3.** PCGF1 regulates local H2AK119ub1 levels in hematopoietic stem and progenitor cells (HSPCs). (**A**) K-means clustering of 3×FLAG-PCGF1, H2AK119ub1, and H3K27me3 ChIP peaks around transcription start site (TSS) (±8.0 kb) of PRC1 target genes. The average levels of chromatin immunoprecipitation (ChIP) peaks in each cluster are plotted in upper columns. (**B**) Box-and-whisker plots showing H2AK119ub1, H3K27me3, and transcription levels of genes in PRC1 targets (clusters 1 and 2) and non-PRC1 targets in control and *Pcgf1^Δ/Δ^* LSK cells. Boxes represent 25–75 percentile ranges. The whiskers represent 10–90 percentile ranges. Horizontal bars represent medians. Mean values are indicated by '+'. **p<0.01; ***p<0.001 by the Student's *t*-test. (**C**) Proportion of bivalent genes in PRC1 targets (clusters 1 and 2) and non-PRC1 targets in LSK cells. Bivalent genes were defined using our previous ChIP-seq data of wild-type LSK cells (*Aoyama et al., 2018*). (**D**) Scatter plots showing the correlation of the fold enrichment values

*Figure 3 continued on next page*

*Figure 3 continued*

against the input signals (ChIP/Input) (TSS ± 2 kb) of H2AK119ub1 between control and *Pcgf1*^Δ/Δ LSK cells. PRC1 targets are indicated in a green box. (**E**) Snapshots of Flag-PCGF1 and H2AK119ub1 ChIP signals at the *Cebpa* and *Cebpe* gene loci. (**F**) ChIP qPCR assays for H2AK119ub1 at the *Cebpa* promoter in control and *Pcgf1*^Δ/Δ LSK cells. The relative amounts of immunoprecipitated DNA are depicted as a percentage of input DNA. Data are shown as the mean ± SEM (n = 3). ***p<0.001 by the Student's *t*-test. (**G**) Box-and-whisker plots showing ATAC signal levels at proximal promoters (TSS ± 2 kb) in *Pcgf1*^Δ/Δ CD135⁻LSK cells relative to those in control cells. The data of all ATAC peaks (n = 18,417), PCGF1 target genes (TG) (n = 670), and non-PCGF1 TG (n = 17,747) are shown. Boxes represent 25–75 percentile ranges. The whiskers represent 10–90 percentile ranges. Horizontal bars represent medians. Mean values are indicated by red crosses. ***p<0.001 by the one-way ANOVA. (**H**) Top DNA motifs identified in ATAC peaks at proximal promoters (TSS ± 2 kb) positively or negatively enriched in *Pcgf1*^Δ/Δ CD135⁻LSK cells compared to corresponding controls.

The online version of this article includes the following source data for figure 3:

**Source data 1.** Raw data for *Figure 3*.

day 28 without efficient production of PB myeloid cells, leading to the accumulation of excess GMPs (*Figure 4C*).

GMPs can be divided into steady-state GMPs (ssGMP) and self-renewing GMPs (srGMP), the latter of which transiently increase during regeneration (*Hérault et al., 2017*). We performed RNA-seq analysis of GMPs isolated from 5-FU-treated WT mice at various time points (*Figure 4—figure supplement 1B and C*), and defined differentially expressed genes (DEGs) on day 8 after 5-FU treatment since srGMPs reportedly most expand on that day (*Figure 4—figure supplement 1B and C*, *Supplementary file 4*; *Hérault et al., 2017*). Of note, a significant portion of upregulated DEGs in day 8 5-FU-treated GMPs were also upregulated in *Pcgf1*^Δ/Δ GMPs (*Figure 4D* and *Supplementary file 4*). These overlapping genes included *Hoxa9*, a PRC1.1 target (*Figure 4—figure supplement 1D*; *Ross et al., 2012*; *Shinoda et al., 2022*; *Tara et al., 2018*), which is highly expressed in srGMPs (*Hérault et al., 2017*). RT-qPCR confirmed significantly higher expression of *Hoxa9* in *Pcgf1*^Δ/Δ GMPs than control GMPs (*Figure 4E*). Correspondingly, 5-FU treatment transiently decreased H2AK119ub1 levels at *Hoxa9* locus around day 10 (*Figure 4F*). ChIP-seq analysis also revealed significant reductions in H2AK119ub1 levels at promoters of upregulated DEGs in day 8 5-FU-treated GMPs in *Pcgf1*^Δ/Δ GMPs (*Figure 4G*). These results suggest that transient inhibition of PRC1.1 de-represses genes critical to expand srGMP during myeloid regeneration, although the expression of PRC1.1 genes remained largely unchanged during regeneration (*Figure 4—figure supplement 1E*).

To better understand the role of PRC1.1 in myeloid progenitors, we performed single-cell RNA-seq (scRNA-seq) of Lin⁻Sca-1⁻c-Kit⁺ myeloid progenitors from control and *Pcgf1*^Δ/Δ mice at steady state. We used data from 6171 control and 6198 *Pcgf1*^Δ/Δ single cells and identified 10 major clusters based on dimension reduction by UMAP (*Figure 4H*). Functional annotation of respective UMAP clusters using previously reported myeloid progenitor cell gene expression profiles (*Nestorowa et al., 2016*) assigned clusters 0, 2, and 4 to GMPs (*Figure 4H*). PCA subdivided GMPs into two major groups (*Figure 4I*). These two groups exhibited distinct expression profiles of *Hoxa9*, *Irf8*, *Csf1r*, and *Il6ra*, key genes differentially expressed between ssGMPs (*Hoxa9*^lo, *Irf8*^hi, *Csf1r*^hi, *Il6ra*^hi) and srGMPs (*Hoxa9*^hi, *Irf8*^lo, *Csf1r*^lo, *Il6ra*^lo) (*Hérault et al., 2017*), and we classified clusters 2 and 4 as ssGMPs and cluster 0 as srGMPs (*Figure 4J*). Gene Ontology and pathway enrichment analyses using DEGs (cluster 2 or 4 versus cluster 0) revealed that clusters 2 and 4 represented more mature myeloid cell populations than cluster 0 (*Figure 4K*). As expected, *Pcgf1*^Δ/Δ myeloid progenitors had a greater proportion of total GMPs including srGMPs than controls (*Figure 4L*). Of note, the frequency of srGMPs was also increased in *Pcgf1*^Δ/Δ myeloid progenitors (*Figure 4L*). These results indicate that PRC1.1 restricts expansion of self-renewing GMPs and suggest that transient PRC1.1 inhibition allows for temporal amplification of GMPs and their subsequent differentiation to mature myeloid cells.

## PCGF1 restricts GMP self-renewal network

To further investigate the mechanism by which PCGF1 regulates GMPs, we took advantage of in vitro culture experiments. Remarkably, *Pcgf1*^Δ/Δ GMPs displayed better proliferation than control GMPs under myeloid-expanding culture conditions (*Figure 5A*). Moreover, while comparable numbers of cells expressing immunophenotypic GMP markers (CD34⁺FcγR⁺c-Kit⁺Sca-1⁻Lineage⁻) (immunophenotypic GMPs) were produced by control and *Pcgf1*^Δ/Δ HSCs on day 7 of culture, control HSCs showed a rapid decline in immunophenotypic GMP production afterward but *Pcgf1*^Δ/Δ HSCs persistently produced immunophenotypic GMPs until day 23 (*Figure 5B*). Of interest, differentiation of the

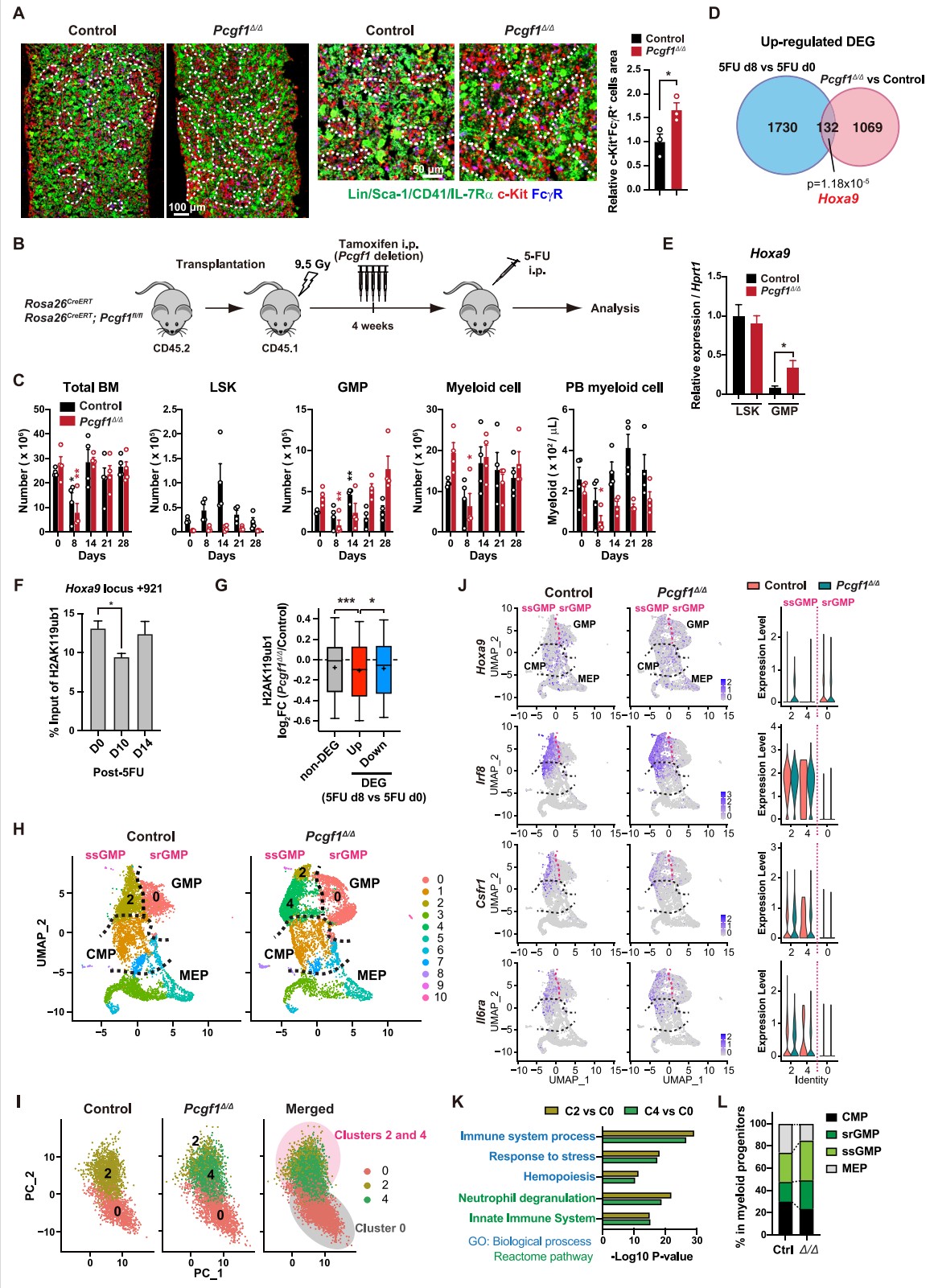

**Figure 4.** PCGF1 negatively regulates granulocyte-macrophage progenitor (GMP) self-renewal. (**A**) Immunofluorescence staining of bone marrow (BM) sections from control and *Pcgf1ᐃ/ᐃ* mice. Magnified images are depicted in the middle panels. Dotted lines denote clusters of GMP (Lin⁻Sca-1⁻CD41⁻IL-7Rα⁻c-Kit⁺FcγR⁺) (c-Kit, red; FcγR, blue; merged, purple). The right panel shows c-Kit⁺FcγR⁺ purple cell area in BM sections relative to that in control. Data are shown as the mean ± SEM (n = 3). (**B**) Strategy to analyze emergency myelopoiesis induced by one shot of 5-FU (150 mg/kg). (**C**) Absolute

*Figure 4 continued on next page*

*Figure 4 continued*

numbers of total BM cells, LSK cells, GMPs, and Mac-1⁺ myeloid cells in a unilateral pair of femur and tibia and Mac-1⁺ myeloid cells in peripheral blood (PB) at the indicated time points post-5-FU injection. Data are shown as the mean ± SEM (n = 4 ; * versus day 0 for each genotype). Circles represent the data from individual mice. (**D**) A Venn diagram showing the overlap between upregulated differentially expressed genes (DEGs) in 5-FU treated GMPs on day 8 and upregulated DEGs in *Pcgf1^{Δ/Δ}* GMPs. (**E**) Quantitative RT-PCR analysis of *Hoxa9* in LSK cells and GMPs. *Hprt1* was used to normalize the amount of input RNA. Data are shown as the mean ± SEM (n = 3). (**F**) Chromatin immunoprecipitation (ChIP) qPCR assays for H2AK119ub1 at the *Hoxa9* locus in GMPs from WT mice on days 0, 10, and 14 post-5-FU treatment. The relative amounts of immunoprecipitated DNA are depicted as a percentage of input DNA. Data are shown as the mean ± SEM (n = 3). (**G**) Fold changes in H2AK119ub1 levels in *Pcgf1^{Δ/Δ}* GMPs relative to control GMPs at the promoters of non-DEGs and DEGs up- and downregulated in day 8 GMPs compared to day 0 GMPs post-5-FU treatment. (**H**) UMAP plots illustrating the identification of cell clusters based on single-cell transcriptomic profiling of control and *Pcgf1^{Δ/Δ}* myeloid progenitors (Lin⁻Sca-1⁻c-Kit⁺). (**I**) Principal component analysis (PCA) plots of control and *Pcgf1^{Δ/Δ}* GMPs individually and in combination. (**J**) UMAP and violin plots showing expression of *Hoxa9*, *Irf8*, *Csf1r*, and *Il-6ra* in control and *Pcgf1^{Δ/Δ}* myeloid progenitors. (**K**) Gene Ontology and pathway enrichment analyses using DEGs in the indicated clusters. (**L**) Proportion of common myeloid progenitors (CMPs), megakaryocyte-erythroid progenitors (MEPs), steady-state GMPs (ssGMPs), and self-renewing GMPs (srGMPs) in control and *Pcgf1^{Δ/Δ}* myeloid progenitors. *p<0.05; **p<0.01; ***p<0.001 by the Student's *t*-test.

The online version of this article includes the following source data and figure supplement(s) for figure 4:

**Source data 1.** Raw data for *Figure 4*.

**Figure supplement 1.** Granulocyte-macrophage progenitor (GMP) expansion during regeneration.

**Figure supplement 1—source data 1.** Raw data for *Figure 4—figure supplement 1*.

expanded *Pcgf1^{Δ/Δ}* GMPs was largely blocked, as indicated by reduced Mac1⁺ differentiated myeloid cells/GMP ratios until day 19 (*Figure 5—figure supplement 1A*), suggesting that *Pcgf1^{Δ/Δ}* GMPs underwent enhanced self-renewal rather than differentiation. However, *Pcgf1^{Δ/Δ}* GMPs did not lose its differentiation potential to mature myeloid cells and did not show evident differentiation block, with more Mac1⁺ cells ultimately generated from HSC cultures than control GMPs (*Figure 5—figure supplement 1A*). This was also consistent that the number of mature myeloid cells expressing myeloid differentiation markers increased in *Pcgf1^{Δ/Δ}* BM and spleen (*Figure 1—figure supplement 2C and D*). Furthermore, *Pcgf1^{Δ/Δ}* HSPCs showed remarkably sustained colony formation activity upon serial replating with myeloid cytokines, which is in line with the elevated self-renewing activity of *Pcgf1^{Δ/Δ}* GMPs (*Figure 5C*).

srGMPs have increased levels of nuclear β-catenin, which is known to confer aberrant self-renewal features to leukemic GMPs (*Wang et al., 2010*) and directly suppresses *Irf8* expression (*Hérault et al., 2017*). The proportion of nuclear β-catenin was significantly increased in *Pcgf1^{Δ/Δ}* GMPs at later time points of culture when GMPs in control culture shrunk but GMPs in *Pcgf1^{Δ/Δ}* culture kept expanding (*Figure 5B and D*). *Pcgf1^{Δ/Δ}* GMPs in culture possessed a transcriptional profile typical to srGMPs; upregulation of *Hoxa9* and downregulation of *Irf8*, *Csf1r*, and *Il6ra* (*Figure 5E*). GSEA revealed activation of the Wnt/β-catenin pathway (*Shooshtarizadeh et al., 2019*) and downregulation of the Irf8 targets (*Kubosaki et al., 2010*) in *Pcgf1^{Δ/Δ}* GMPs (*Figure 5F*). We hypothesized that *Hoxa9*, a direct target of PCGF1, could have a role in the GMP self-renewal network. Overexpression of *Hoxa9* in HSPCs significantly enhanced their growth and induced persistent production of GMPs for a long period (*Figure 5G* and *Figure 5—figure supplement 1B*). Most *Hoxa9*-overexpressing GMPs had nuclear β-catenin (*Figure 5H*). These results indicate that HoxaA9 can reinforce activation of β-catenin, thus placing Hoxa9 as a component of the GMP self-renewal network and PCGF1-PRC1.1 as a negative regulator of this network (*Figure 5I*). The promoter region of the *Ctnnb1* gene contains at least one predicted sequence for Hoxa9 binding (CTTATAAATCG) (data not shown). We performed qPCR analysis using *Hoxa9* overexpression samples at day 12 of culture, in which nuclear localization of β-catenin was observed (*Figure 5H*). *Ctnnb1* gene expression was slightly but significantly upregulated in *Hoxa9* overexpression samples compared to mock control (*Figure 5—figure supplement 1C*). These results raise the possibility that Hoxa9 is a direct transcriptional activator of the *Ctnnb1* gene, but this warrants further analysis.

## Constitutive PCGF1 loss promotes malignant transformation

In leukemia, GMP clusters are constantly produced owing to persistent activation of the myeloid self-renewal network and a lack of termination cytokines that normally restore HSC quiescence (*Hérault et al., 2017*). A significant portion of *Pcgf1^{Δ/Δ}* mice, which exhibit constant production of GMP clusters, developed lethal myeloproliferative neoplasms (MPN) with severe anemia and massive

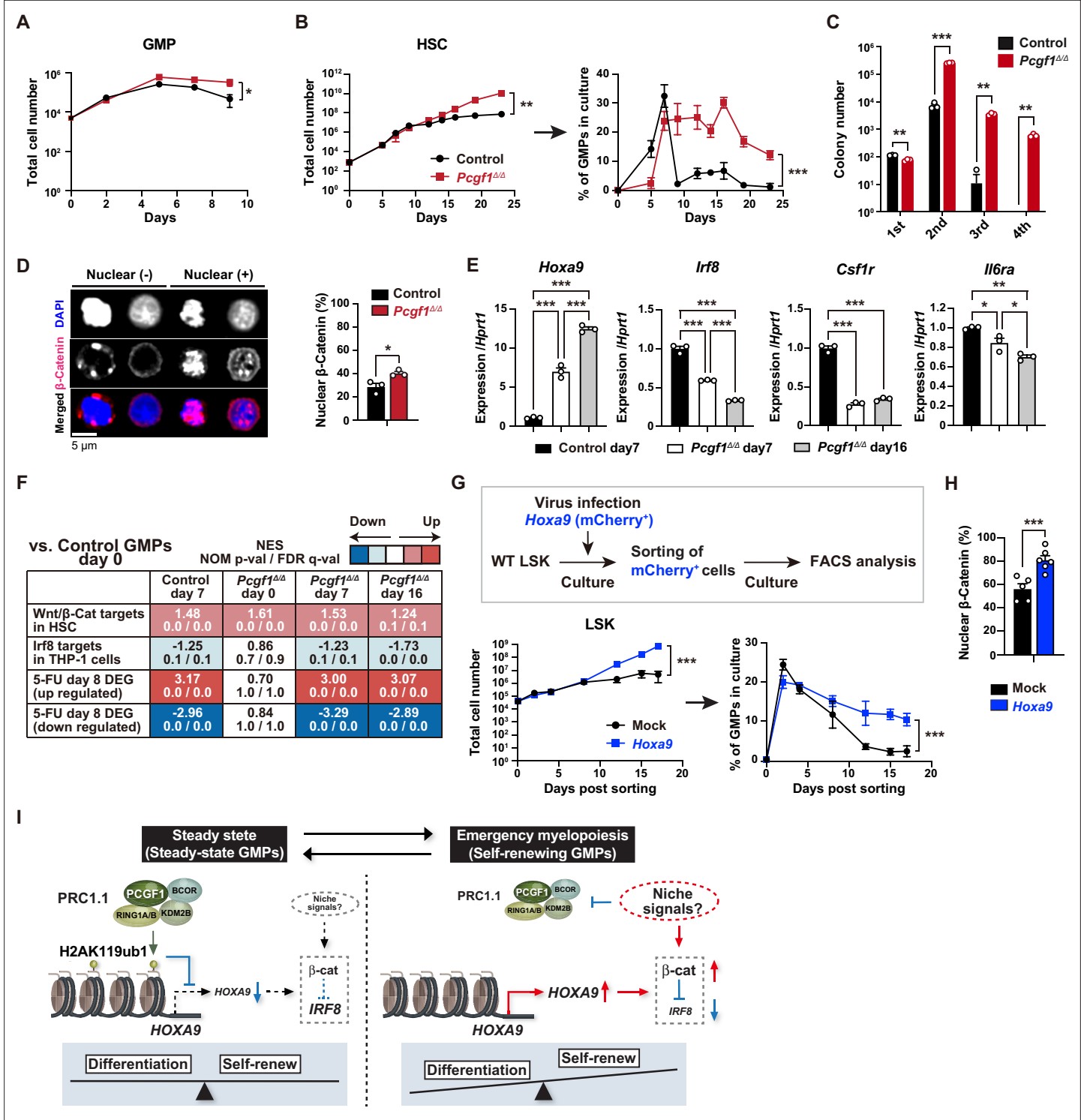

**Figure 5.** PCGF1 restricts self-renewal of granulocyte-macrophage progenitors (GMPs) by attenuating *Hoxa9* expression. (**A**) Growth of control and *Pcgf1*^Δ/Δ^ GMPs in culture. Cells were cultured in triplicate under myeloid culture condition-2 (25 ng/mL SCF, TPO, Flt3L, and IL-11 and 10 ng/mL IL-3 and GM-CSF). Data are shown as the mean ± SD (n = 3). (**B**) Growth of control and *Pcgf1*^Δ/Δ^ hematopoietic stem cells (HSCs) under myeloid culture condition-2 (25 ng/mL SCF, TPO, Flt3L, and IL-11 and 10 ng/mL IL-3 and GM-CSF). Cells were cultured in triplicate. The proportion of GMPs in culture is depicted on the right panel. Data are shown as the mean ± SD (n = 3). (**C**) Replating assay data. 700 LSK cells were plated in a methylcellulose medium containing 20 ng/mL of SCF, TPO, IL-3, and GM-CSF. After 10 d of culture, colonies were counted and pooled, and 1 × 10^4^ cells were then replated in the same medium every 7 d. Data are shown as the mean ± SEM (n = 3). (**D**) Proportion of immunophenotypic GMPs with nuclear β-catenin in control and *Pcgf1*^Δ/Δ^ immunophenotypic GMPs in HSC culture on day 16 in (**B**). Representative immunofluorescent signals of β-catenin in control

*Figure 5 continued on next page*

*Figure 5 continued*

immunophenotypic GMPs are shown on the right panel. Data are shown as the mean ± SEM (n = 3). (**E**) Quantitative RT-PCR analysis of *Hoxa9*, *Irf8*, *Csf1r*, and *Il-6ra* in sorted control and *Pcgf1^(Δ/Δ)^* immunophenotypic GMPs in HSC culture in (**B**) at the indicated time points. *Hprt1* was used to normalize the amount of input RNA. Data are shown as the mean ± SEM (n = 3). (**F**) Gene set enrichment analysis (GSEA) using RNA-seq data. The gene sets used are indicated in ***Supplementary file 1***. (**G**) Growth of mock control and *Hoxa9*-expressing LSK cells. LSK cells transduced with a *Hoxa9* retrovirus harboring mCherry marker gene were cultured in triplicate under myeloid culture condition-2 (25 ng/mL SCF, TPO, Flt3L, and IL-11 and 10 ng/mL IL-3 and GM-CSF). The proportion of GMPs in culture is depicted on the right panel. Data are shown as the mean ± SD (n = 4). (**H**) Proportion of GMPs with nuclear β-catenin in mock control and *Hoxa9*-expressing GMPs in LSK culture on day 12 in (**G**). Data are shown as the mean ± SEM (n = 5–6). (**I**) Model of the molecular network controlling GMP self-renewal and differentiation. *p<0.05; **p<0.01; ***p<0.001 by the Student's *t*-test (**A–D**, **H**, and **G**) or the one-way ANOVA (**E**). Each symbol is derived from an individual culture.

The online version of this article includes the following source data and figure supplement(s) for figure 5:

**Source data 1.** Raw data for *Figure 5*.

**Figure supplement 1.** Granulocyte-macrophage progenitor (GMP) expansion in the absence of PCGF1.

**Figure supplement 1—source data 1.** Raw data for *Figure 5—figure supplement 1*.

accumulation of mature myeloid cells in PB, BM, and spleen (***Figure 6A–E***). Morphological analysis of BM and spleen sections revealed accumulation of mature myeloid cells but no obvious differentiation block like acute myeloid leukemia (AML) was observed (***Figure 6F***). Indeed, AML was not observed in long-term observation of *Pcgf1^(Δ/Δ)^* mice. It is assumed that the epigenetic status that allows GMP self-renewal was enforced over time in the absence of *Pcgf1*, leading to enhanced production of mature myeloid cells that mimic MPN. A part of *Pcgf1^(Δ/Δ)^* mice also developed lethal T-cell acute lymphoblastic leukemia (T-ALL) (***Figure 6A and G***) like *Bcor* mutant mice (***Tara et al., 2018***). These results indicate that constitutive activation of the GMP self-renewal network in *Pcgf1^(Δ/Δ)^* mice serves to promote malignant transformation. Taken together, these results highlight the importance of PRC1.1-dependent suppression of the myeloid self-renewing network to prevent malignant transformation.

## Discussion

In this study, we demonstrated that PCGF1 contributes to balanced hematopoiesis by restricting precocious myeloid commitment of HSPCs and expansion of myeloid progenitors while its inhibition promotes emergency myelopoiesis and myeloid transformation. These findings present a sharp contrast with PCGF4/BMI1 essential for self-renewal of HSCs (***Iwama et al., 2004***; ***Oguro et al., 2006***; ***Park et al., 2003***) and underscore distinct functions between canonical PRC1 and non-canonical PRC1.1 in hematopoiesis (***Figure 6H***).

PcG and trithorax group proteins mark developmental regulator gene promoters with bivalent histone domains to keep them poised for activation in ES cells (***Bernstein et al., 2006***). We previously reported that canonical PRC1 reinforces bivalent domains at the B cell regulator genes, *Ebf1* and *Pax5*, to maintain B cell lineage commitment poised for activation in HSPCs (***Oguro et al., 2010***). In contrast, PCGF1 appeared to target non-bivalent PRC1 target genes marked with moderate levels of H2AK119ub1 and H3K27me3. Among these, PCGF1 targets myeloid regulator genes, such as *Cebpa*, thereby negatively regulating myeloid commitment. Our findings indicate that canonical and non-canonical PRC1 restrict the lymphoid and myeloid commitment of HSPCs, respectively, by targeting different transcriptional programs of differentiation, thereby fine-tuning the balance of HSPC commitment (***Figure 6H***). Although there might be considerable functional redundancy between canonical and non-canonical PRC1 complexes, our results uncovered a unique function of PRC1.1 in the lineage commitment of HSPCs. The reduction in the levels of *Cebpa* expression by introducing a *Cebpa^(fl/+)^* allele in *Pcgf1^(Δ/Δ)^* LSK cells was sufficient to restore balanced production of myeloid and B cells by *Pcgf1^(Δ/Δ)^* HSPCs in culture (***Figure 2E***). However, we could not confirm these data in mice probably due to inefficient deletion of *Cebpa* using a Cre-ERT system (data not shown). The real impact of de-repressed *Cebpa* in myeloid-biased differentiation of *Pcgf1^(Δ/Δ)^* HSPCs requires further validation in vivo.

Myeloid-biased output from HSPCs is one of the hallmarks of emergency hematopoiesis (***Trumpp et al., 2010***; ***Zhao and Baltimore, 2015***). In mouse models of regeneration, myeloid-biased MPP2 and MPP3 are transiently overproduced, suggesting that HSCs produce functionally distinct lineage-biased MPPs to adapt blood production to hematopoietic demands (***Pietras et al., 2015***). In the present study, we found that *Pcgf1*-deficient hematopoiesis recapitulates sustained emergency myelopoiesis,

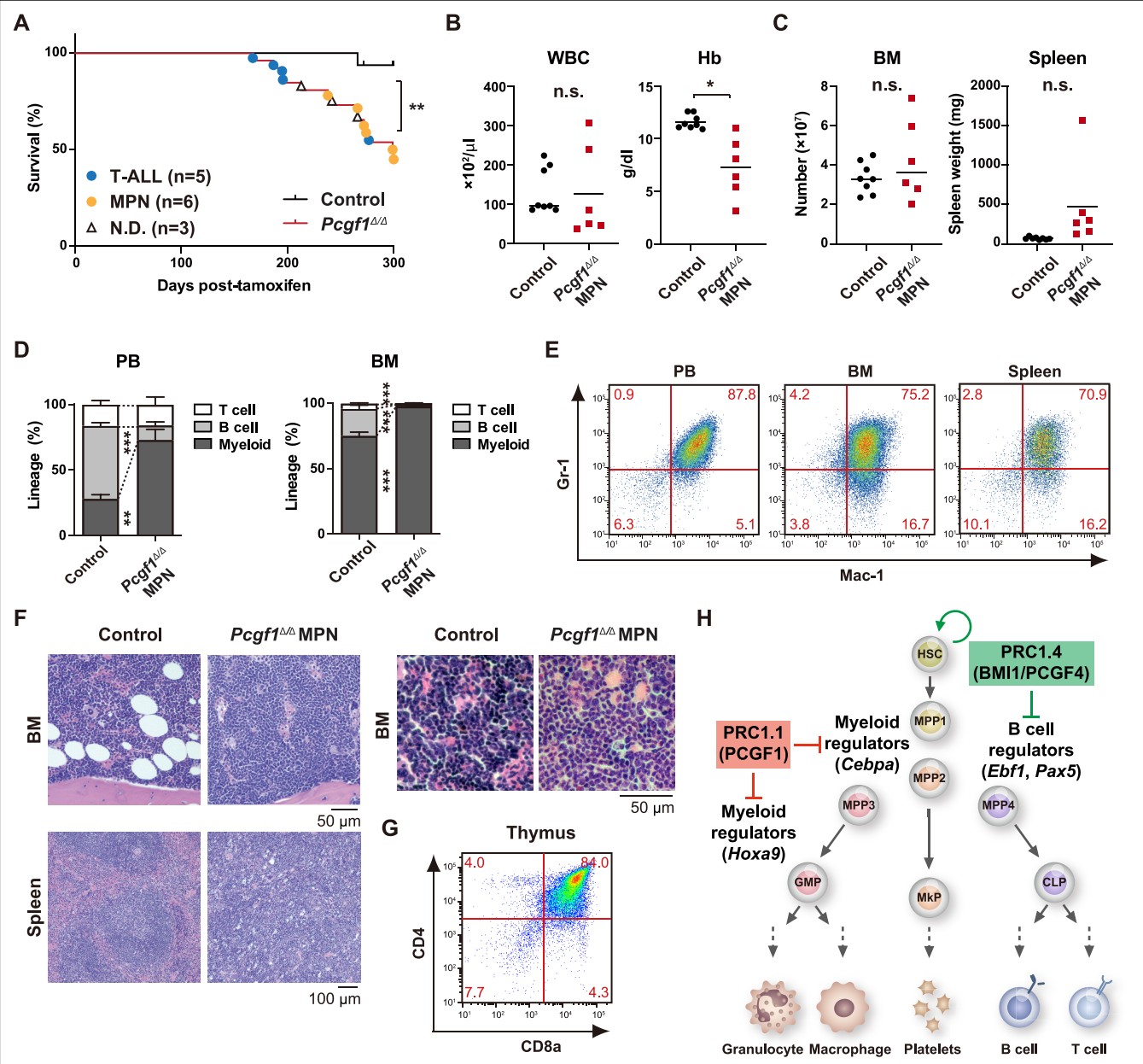

**Figure 6.** Development of lethal myeloproliferative neoplasm in *Pcgf1*^Δ/Δ mice. (**A**) Kaplan–Meier survival curves of control (n = 7) and *Pcgf1*^Δ/Δ (n = 25) mice after the tamoxifen injection. (**B**) White blood cell (WBC) and hemoglobin (Hb) in peripheral blood (PB) from control (n = 8) and moribund *Pcgf1*^Δ/Δ myeloproliferative neoplasms (MPN) mice (n = 6). Bars indicate median values. (**C**) Absolute numbers of total bone marrow (BM) cells and spleen weight in control (n = 8) and moribund *Pcgf1*^Δ/Δ MPN mice (n = 6). (**D**) The proportions of Mac-1+ and/or Gr-1+ myeloid cells, B220+ B cells, and CD4+ or CD8+ T cells in PB and BM in control (n = 8) and moribund *Pcgf1*^Δ/Δ MPN mice (n = 6). Data are shown as the mean ± SEM. (**E**) Representative flow cytometric profiles of PB, BM, and spleen of control and moribund *Pcgf1*^Δ/Δ MPN mice. The percentages of gated populations over CD45.2+ live cells are indicated. (**F**) Representative histology of BM and spleen from control and moribund *Pcgf1*^Δ/Δ MPN mice observed by hematoxylin-eosin staining (left). The high-power field images of BM from control and moribund *Pcgf1*^Δ/Δ MPN mice observed by hematoxylin-eosin staining (right). (**G**) Representative flow cytometric profiles of thymus from control mice and moribund *Pcgf1*^Δ/Δ T-ALL mice. (**H**) Model for the stage-specific roles of non-canonical PRC1 in hematopoietic differentiation. *p<0.05; **p<0.01; ***p<0.001 by the Student's *t*-test. Each symbol is derived from an individual mouse.

The online version of this article includes the following source data for figure 6:

**Source data 1.** Raw data for *Figure 6*.

although the production of circulating myeloid cells was not enhanced. Expanding GMPs, GMP clusters during regeneration, which, in turn, differentiate into granulocytes (*Hérault et al., 2017*). Of note, PCGF1 loss induced constitutive GMP cluster formation at steady state and sustained GMP expansion in mice after myeloablation and in culture. Correspondingly, *Pcgf1*-deficient mice had a greater number of self-renewing GMPs than control mice. This unique phenotype may implicate the importance of transient but not constitutive PCGF1 repression for proper myeloid regeneration. β-catenin and *Irf8* constitute an inducible self-renewal progenitor network controlling GMP cluster formation, with β-catenin directly suppressing *Irf8* expression while restoration of *Irf8* expression terminating the self-renewal network and inducing GMP differentiation (*Hérault et al., 2017*; *Figure 5I*). *Hoxa9*, which is upregulated in srGMPs, is one of the PRC1.1 targets in myeloid progenitors (*Figure 4—figure supplement 1D*; *Ross et al., 2012*; *Shinoda et al., 2022*; *Tara et al., 2018*). We demonstrated that *Hoxa9* expression activates β-catenin and promotes GMP self-renewal, identifying Hoxa9 as a component of the GMP self-renewal network. Of note, PRC1.1 is transiently inhibited to de-repress such GMP self-renewal network genes. This transient nature of PRC1.1 inhibition allows for srGMP expansion and GMP cluster formation followed by proper differentiation of expanded GMPs. As expression levels of PRC1.1 components remained unchanged during hematopoietic regeneration, non-canonical PRC1.1 activity could be modulated by post-translational modifications in response to extracellular stimuli like canonical PRC1 (*Banerjee Mustafi et al., 2017*; *Liu et al., 2012*; *Nacerddine et al., 2012*; *Voncken et al., 2005*). How extrinsic signals modulate PRC1.1 functions to regulate myelopoiesis remains an important question.

The molecular machineries that drive emergency myelopoiesis are often hijacked by transformed cells (*Hérault et al., 2017*). A significant portion of *Pcgf1*-deficient mice eventually developed lethal MPN after a sustained myeloproliferative state. These findings indicate that PRC1.1 functions as a critical negative regulator of myeloid transformation. Among the components of PRC1.1, *BCOR* and *BCLRL1*, but not *PCGF1* are targeted by somatic gene mutations in various hematological malignancies, including myelodysplastic syndrome (MDS), chronic myelomonocytic leukemia (CMML), and acute myeloid leukemia (AML) (*Isshiki and Iwama, 2018*). We reported that mice expressing a carboxyl-terminal truncated BCOR, which cannot interact with PCGF1, showed myeloid-biased hematopoiesis like *Pcgf1*-deficient mice. Importantly, HSPCs in these mice showed a growth advantage in the myeloid compartment, which was further enhanced by the concurrent deletion of *Tet2*, leading to the development of lethal MDS (*Tara et al., 2018*). De-repression of myeloid regulator genes, such as *Cebp* family and *Hoxa* cluster genes, were also detected in *Bcor* mutant progenitor cells (*Tara et al., 2018*). These findings also support the idea that PRC1.1 restricts myeloid transformation by transcriptionally repressing aberrant activation of myeloid regeneration programs.

Collectively, our findings highlight a critical role of PRC1.1 in coordinating steady-state and emergency hematopoiesis and preventing malignant transformation. They also suggest that transient inhibition of PRC1.1 would be a novel approach to temporarily induce emergency myelopoiesis and enhance myeloid cell supply while avoiding the potential risk for malignant transformation.

## Materials and methods

### Mice

Wild-type mice (C57BL/6) and *Rosa::Cre-ERT2* mice were purchased from the Japan SLC and TaconicArtemis GmbH, respectively. *Pcgf1*^fl^ and *Cebpa*^fl^ mice were kindly provided by Haruhiko Koseki and Daniel G. Tenen, respectively, and previously reported (*Almeida et al., 2017*; *Zhang et al., 2004*). All experiments using mice were performed in accordance with our institutional guidelines for the use of laboratory animals and approved by the Review Board for Animal Experiments of Chiba University (approval ID: 30-56) and the University of Tokyo (approval ID: PA18-03).

### Bone marrow transplantation

To generate hematopoietic cell-specific *Pcgf1* KO mice, we transplanted total BM cells ($5 \times 10^6$) from *Rosa26*^CreERT^ and *Rosa26*^CreERT^;*Pcgf1*^fl/fl^ mice into lethally irradiated (9.5 Gy) CD45.1 recipient mice. For competitive bone marrow transplantation assay, we transplanted total BM cells ($2 \times 10^6$) from CD45.2 donor mice with CD45.1+ competitor total BM cells ($2 \times 10^6$) into lethally irradiated (9.5 Gy) CD45.1 recipient mice. To induce Cre activity, transplanted mice were injected with 100 µL of tamoxifen

(Sigma-Aldrich) dissolved in corn oil (Sigma-Aldrich) at a concentration of 10 mg/mL intraperitoneally once a day for five consecutive days 4 wk after transplantation.

## Locus-specific genotyping of *Pcgf1*, *Cebpa*, and *Rosa::Cre-ERT*

To detect $Pcgf1^{fl}$, $Pcgf1^{\Delta}$, $Cebpa^{fl}$, $Cebpa^{\Delta}$, and *Rosa::Cre-ERT* PCR reactions were performed using the specific oligonucleotides. The oligonucleotide sequences used are shown in *Supplementary file 5*.

## 5-FU challenge

8–12-week-old wild-type mice or control and $Pcgf1^{\Delta/\Delta}$ were injected with 300 μL PBS or 150 mg/kg (3.75 mg per 25 g body weight mouse) 5-FU (Kyowa KIRIN) dissolved in 300 μL PBS intraperitoneally once.

## Flow cytometry analyses and antibodies

The monoclonal antibodies recognizing the following antigens were used in flow cytometry and cell sorting: CD45.1 (A20), CD45.2 (104), Gr-1 (RB6-8C5), CD11b/Mac-1 (M1/70), Ter-119 (TER-119), B220 (RA3-6B2), CD127/IL-7R (SB/119), CD4 (GK1.5), CD8a (53–6.7), CD117/c-Kit (2B8), Sca-1 (D7), CD34 (RAM34), CD150 (TC15-12F12.2), CD48 (HM48-1), CD135 (A2F10), CD16/32/FcγRII-III (93), CD41 (eBioMWReg30), CD105 (MJ7/18), Ly6D (49-H4), Ly6G (HK1.4), Ly6G (1A8), F4/80 (BM8), lineage mixture (Gr-1, Mac-1, Ter-119, CD127/IL-7R, B220, CD4, CD8α), and lineage mixture for CLP (Gr-1, Mac-1, Ter-119, B220, CD4, CD8α). Monoclonal antibodies were purchased from BioLegend, Tonbo Biosciences, Thermo Fisher Scientific, or BD Biosciences. Dead cells were eliminated by staining with 0.5 μg/mL propidium iodide (Sigma-Aldrich). All flow cytometric analyses and cell sorting were performed on FACSAria IIIu, FACSCanto II, and FACSCelesta (BD Biosciences). Cell surface protein expression used to define hematopoietic cell types were as follows:

> HSC: $CD150^+CD48^-CD135^-CD34^-c\text{-}Kit^+Sca\text{-}1^+Lineage^-$
> MPP1: $CD150^+CD48^-CD135^-CD34^+c\text{-}Kit^+Sca\text{-}1^+Lineage^-$
> MPP2: $CD150^+CD48^+CD135^-CD34^+c\text{-}Kit^+Sca\text{-}1^+Lineage^-$
> MPP3: $CD150^-CD48^+CD135^-CD34^+c\text{-}Kit^+Sca\text{-}1^+Lineage^-$
> MPP4: $CD150^-CD48^+CD135^+CD34^+c\text{-}Kit^+Sca\text{-}1^+Lineage^-$
> CMP: $CD34^+FcγR^-c\text{-}Kit^+Sca\text{-}1^-Lineage^-$
> GMP: $CD34^+FcγR^+c\text{-}Kit^+Sca\text{-}1^-Lineage^-$
> MEP: $CD34^-FcγR^-c\text{-}Kit^+Sca\text{-}1^-Lineage^-$
> pre-GM: $CD150^-CD105^-FcγR^-CD41^-c\text{-}Kit^+Sca\text{-}1^-Lineage^-$
> CLP: $c\text{-}Kit^{low}Sca\text{-}1^{low}CD135^+IL7R^+Lineage\ (for\ CLP)^-$
> ALP: $Ly6D^-cKit^{low}Sca\text{-}1^{low}CD135^+IL7R^+Lineage^-$
> BLP: $Ly6D^+cKit^{low}Sca\text{-}1^{low}CD135^+IL7R^+Lineage^-$
> LSK: $c\text{-}Kit^+Sca\text{-}1^+Lineage^-$
> LK: $c\text{-}Kit^+Lineage^-$
> Pro-B: $B220^+CD43^+IgM^-$
> Pre-B: $B220^+CD43^-IgM^-$

## Quantitative RT-PCR

Total RNA was extracted using a RNeasy Micro Plus Kit (QIAGEN) or TRIZOL LS solution (MOR) and reverse transcribed by the SuperScript IV First-Strand Synthesis System (Invitrogen) or the ReverTra Ace α- (TOYOBO) with an oligo-dT primer. Real-time quantitative PCR was performed with a StepOnePlus Real-Time PCR System (Life Technologies) using FastStart Universal Probe Master (Roche) and the indicated combinations of the Universal Probe Library (Roche), or TB Green Premix Ex Taq II (TaKaRa Bio). All data are presented as relative expression levels normalized to *Hprt* expression. The primer sequences used are shown in *Supplementary file 5* (*Murakami et al., 2021*; *Sonntag et al., 2018*).

## Limiting dilution assay

For in vivo limiting dilution assay, we transplanted limiting numbers of total BM cells ($1 \times 10^4$, $4 \times 10^4$, $8 \times 10^4$, and $2 \times 10^5$) isolated from primary recipients (control and $Pcgf1^{\Delta/\Delta}$ mice 1 mo after tamoxifen

injections) with CD45.1$^+$ competitor total BM cells (2 × 10$^5$) into lethally irradiated CD45.1 recipient mice. PB analyses were performed at 16 wk after transplantation.

For in vitro limiting dilution assay, we sorted HSCs from control and *Pcgf1*$^{\Delta/\Delta}$ mice 1 mo after tamoxifen injections and cultured limiting numbers of the cells (1, 5, 25, and 125) with TSt-4 (B cells) or TSt-4/DLL1 stromal cells (T cells) in RPMI (Thermo Fisher Scientific) supplemented with 10% BSA (093001; STEMCELL Technologies), 50 μM 2-ME (Sigma-Aldrich), 100 μM MEM Non-Essential Amino Acids solution (Gibco), 100 μM sodium pyruvate (Gibco), and 2 ng/mL recombinant mouse IL-7 (577802; BioLegend) for 28 d. The generation of CD19$^+$ B cells or Thy1.2$^+$ T cells in each well was detected by flow cytometry.

## Cell cycle assay

BM cells were stained with antibodies against cell-surface markers. After washing, cells were fixed and permeabilized with a BD Phosflow Lyse/Fix Buffer and a BD Phosflow Perm Buffer II (BD Biosciences) according to the manufacturer's instructions. Cells were stained with FITC-Ki67 antibody (#11-5698-82; Thermo Fisher Scientific) at room temperature for 30 min and then with 1 μg/mL 7-AAD (Sigma-Aldrich). Flow cytometric analyses were performed on FACSAria IIIu (BD Biosciences).

## Cell culture

For growth assays, sorted CD34$^-$CD150$^+$LSK HSCs were cultured in S-Clone SF-O3 (Sanko Junyaku) supplemented with 0.1% BSA (093001; STEMCELL Technologies), 50 μM 2-ME (Sigma-Aldrich) and 1% penicillin/streptomycin/glutamine (Gibco). 20 ng/mL of recombinant mouse SCF (579706; BioLegend) and recombinant human TPO (763706; BioLegend) for HSC culture conditions and 10 ng/mL of SCF, TPO, recombinant mouse IL-3 (575506; BioLegend), and recombinant murine GM-CSF (315-03; PeproTech) for myeloid culture condition-1 were added to cultures. In the case of myeloid culture condition-2, sorted CD150$^+$CD48$^-$CD135$^-$CD34$^-$LSK HSCs, LSK cells, and GMPs were cultured in IMDM (Gibco) supplemented with 5% FBS, 50 μM 2-ME (Sigma-Aldrich), 1% penicillin/streptomycin/glutamine (Gibco), 1 mM sodium pyruvate (Gibco), and 0.1 mM MEM Non-Essential Amino Acids solution (Gibco). 25 ng/mL of SCF, TPO, recombinant human Flt3L (300-19; PeproTech) and recombinant murine IL-11 (220-11; PeproTech) and 10 ng/mL of IL-3 and GM-CSF were added to cultures.

For replating assays, LSK cells were plated in methylcellulose medium (Methocult M3234; STEMCELL Technologies) containing 20 ng/mL of SCF, TPO, IL-3, and GM-CSF.

## Retroviral vector and virus production

Full-length *Pcgf1* and *Bmi1* cDNA tagged with a 3×Flag at the N-terminus was subcloned into the retroviral vector pGCDNsam-IRES-EGFP. Full-length *Hoxa9* cDNA was subcloned into the retroviral vector pMYs-IRES-mCherry. A recombinant retrovirus was generated by a 293gpg packaging cell line. The virus in supernatants of 293gpg cells was concentrated by centrifugation at 6000 × *g* for 16 hr.

## Immunofluorescence imaging of bone marrow and spleen sections

Isolated mouse femurs were immediately placed in ice-cold 2% paraformaldehyde solution (PFA/PBS) and fixed under gentle agitation for 16 hr. The samples were then incubated in 15 and 30% sucrose for cryoprotection overnight. Samples were embedded in O.C.T. (Sakura) and frozen in cooled hexane. The 7 μm frozen sections were generated with a cryostat (Cryostar NX70, Thermo Scientific) using Kawamoto's tape method (*Kawamoto, 2003*). Sections on slide glasses were blocked with staining buffer (10% normal donkey serum in TBS) and an Avidin/Biotin Blocking Kit (VECTOR), then stained with biotinylated anti-lineage antibody cocktail and anti-c-Kit antibody (#AF 1356; R&D Systems), or anti-FcγR-Alexa Fluor 647 (#101314; BioLegend) in staining buffer overnight at 4°C. For secondary staining, sections were incubated with streptavidin-Alexa Fluor 488 (#S11223; Invitrogen) and donkey anti-goat Alexa Fluor 555 (#A21432; Invitrogen) antibody for 3 hr at room temperature. Finally, sections were incubated with 1 μg/mL DAPI/TBS for 10 min and mounted with ProLong Glass Antifade Mountant (Thermo Scientific). Images of sections were captured on a confocal microscope (Dragonfly, Andor, or A1Rsi, Nikon) and processed using Fiji. ImageJ was used for image quantification.

## Immunofluorescence imaging of purified GMPs

GMPs were sorted directly onto glass slides using BD AriaIIIu. The cells were washed three times with PBS for 5 min between each staining step. Cells were fixed with 4% PFA for 15 min, permeabilized

with 0.1% Triton X-100 for 10 min, and then blocked with 3% BSA for 1 hr at room temperature. The cells were then incubated with rabbit anti-mouse β-catenin (#9582S; Cell Signaling) primary antibody at 4°C overnight. The cells were then stained with anti-rabbit AF488A (#20015; Biotium) secondary antibody for 2 hr at room temperature. After staining with 1 µg/mL DAPI/PBS for 5 min, the cells were mounted with ProLong Glass Antifade Mountant (Thermo Fisher). DragonFly (Andor, ×40 objective) was used for image acquisition.

## Bulk RNA-seq and data processing

Total RNAs were extracted from 1000 to 5000 cells using an RNeasy Plus Micro Kit (QIAGEN) and cDNAs were synthesized using a SMART-Seq v4 Ultra Low Input RNA Kit for Sequencing (Clontech) according to the manufacturer's instructions. The ds-cDNAs were fragmented using S220 or M220 Focused-ultrasonicator (Covaris), then cDNA libraries were generated using a NEBNext Ultra DNA Library Prep Kit (New England BioLabs) according to the manufacturer's instructions. Sequencing was performed using HiSeq1500 or HiSeq2500 (Illumina) with a single-read sequencing length of 60 bp. TopHat2 (version 2.0.13; with default parameters) was used to map the reads to the reference genome (UCSC/mm10) with annotation data from iGenomes (Illumina). Levels of gene expression were quantified using Cuffdiff (Cufflinks version 2.2.1; with default parameters). Significant expression differences were detected edgeR (version 3.14; with default parameters), with raw counts generated from String Tie. The super-computing resource was provided by the Human Genome Center, the Institute of Medical Science, the University of Tokyo (http://sc.hgc.jp/shirokane.html). The enrichment analyses were performed using g:Profiler tool.

## Single-cell RNA-seq and data processing

Control ($1.2 \times 10^4$) and $Pcgf1^{\Delta/\Delta}$ ($1.2 \times 10^4$) LK cells were collected for single-cell RNA-seq. mRNA were isolated and libraries were prepared according to Chromium Next GEM Single Cell 3′ Reagent Kits v3.1 (10X Genomics). Raw data files (Base call files) were demultiplexed into fastq files using Cell Ranger with mkfastq command. Then, 'cellranger count' command was used for feature counts, barcode counts with reference 'refdata-gex-mm10-2020-A.' Filtered_feature_bc_matrix included 6565 control LK cells and 7651 $Pcgf1^{\Delta/\Delta}$ LK cells. We subsampled 6565 cells from 7651 $Pcgf1^{\Delta/\Delta}$ LK cells to adjust cell numbers between $Pcgf1^{\Delta/\Delta}$ and control LK. Subsequent analyses were performed using Seurat 4.1.0. Quality filtering for each feature and cell was conducted based on these criteria ( min.cells = 3 & min.features = 200 & nFeature_RNA >200 & nFeature_RNA <10000 & percent.mt <5). After quality filtering, 6171 control LKs and 6198 KO LKs were used for further analysis. Feature counts are log-normalized with the function of 'NormalizeData.' 2000 highly variable features are selected for PCA. PC 1–10 components are used for UMAP and graph-based clustering with the functions of FindNeighbors(object, reduction = "pca", dims = 1:15) and FindClusters(object, resolution = 0.28). Cluster 0, 2, and 4 cells are extracted and reanalyzed with PCA. DEGs are selected with the function of 'FindMarkers(object, min.pct=0.25)'.

## Chromatin immunoprecipitation (ChIP) assays and ChIP-sequencing

ChIP assays for histone modifications were performed as described previously (*Aoyama et al., 2018*) using an anti-H2AK119ub1 (#8240S; Cell Signaling Technology) and an anti-H3K27me3 (#07-449; Millipore). BM LSK cells were fixed with 1% FA at 37°C for 2 min, lysed in ChIP buffer (10 mM Tris-HCl pH 8.0, 200 mM NaCl, 1 mM CaCl₂, 0.5% NP-40 substitute and cOmplete proteases inhibitor cocktail) and sonicated for 5 s ×3 times by a Bioruptor (UCD-300; Cosmo Bio). Then, cells were digested with Micrococcal Nuclease at 37°C for 40 min (New England BioLabs) and added 10 mM EDTA to stop the reaction. After the addition of an equal volume of RIPA buffer (50 mM Tris-HCl pH 8.0, 150 mM NaCl, 2 mM EDTA pH 8.0, 1% NP-40 substitute, 0.5% sodium deoxycholate, 0.1% SDS and cOmplete proteases inhibitor cocktail), cells were sonicated again for 5 s × 10 times by a Bioruptor. After centrifugation, supernatants were immunoprecipitated at 4°C overnight with Dynabeads Sheep Anti-Rabbit IgG (Invitrogen) conjugated with each antibody. Immunoprecipitates were washed with ChIP wash buffer (10 mM Tris-HCl pH 8.0, 500 mM NaCl, 1 mM CaCl₂, 0.5% NP-40 substitute, and cOmplete proteases inhibitor cocktail) four times and TE buffer (10 mM Tris-HCl pH 8.0 and 1 mM EDTA pH 8.0) twice. Bound chromatins and 30 µL of input DNA were suspended in 95 µL and 65 µL elution buffer (50 mM Tris-HCl pH 8.0, 10 mM EDTA pH 8.0, 1% SDS, and 250 mM NaCl), respectively. After the

addition of 5 μL of 5 M NaCl, the solutions were incubated at 65°C for 4 hr, treated with 25 μg/mL RNase A (Sigma-Aldrich) at 37°C for 30 min and 0.1 mg/mL proteinase K (Roche) at 50°C for 1 hr and were purified with a MinElute PCR Purification Kit (QIAGEN).

In 3×Flag-Pcgf1ChIP assay, BM LK cells were fixed with 1% FA at 25°C for 10 min, lysed in RIPA buffer, and sonicated for 11 s ×15 times by a homogenizer (NR-50M; Micro-tec Co.). After centrifugation, supernatants were immunoprecipitated at 4°C overnight with Dynabeads Sheep Anti-Mouse IgG (Invitrogen) conjugated with an anti-FLAG antibody (Sigma-Aldrich). Then, the samples were treated in the same way as ChIP assays for histone modifications.

In ChIP-qPCR assay, quantitative real-time PCR was performed with a StepOnePlus Thermal Cycler (Thermo Fisher Scientific) using SYBR Premix Ex Taq II or TB Green Premix Ex Taq II (Takara Bio). The primer sequences used are shown in *Supplementary file 5*.

ChIP-seq libraries were prepared using a ThruPLEX DNA-seq Kit (Clontech) according to the manufacturer's instructions. Bowtie2 (version 2.2.6; with default parameters) was used to map the reads to the reference genome (UCSC/mm10). The RPM (reads per million mapped reads) values of the sequenced reads were calculated every 1000 bp bin with a shifting size of 100 bp using bedtools. In order to visualize with Integrative Genomics Viewer (IGV) (http://www.broadinstitute.org/igv), the RPM values of the immunoprecipitated samples were normalized by subtracting the RPM values of the input samples in each bin and converted to a bigwig file using wigToBigWig tool. The super-computing resource was provided by the Human Genome Center, the Institute of Medical Science, the University of Tokyo (http://sc.hgc.jp/shirokane.html).

## Assay for transposase accessible chromatin with high-throughput (ATAC)-sequencing

BM CD135⁻LSK cells (1.6–3.0 × $10^4$) and GMPs (3.0 × $10^4$) were lysed in cold lysis buffer (10 mM Tris-HCl pH 7.4, 10 mM NaCl, 3 mM $MgCl_2$, 0.1% IGEPAL CA-630) on ice for 10 min. After centrifugation, nuclei pellets were resuspended with 50 μL of transposase reaction mix (25 μL Tagment DNA buffer (illumine), 2.5 μL Tagment DNA enzyme (illumine) and 22.5 μL water), incubated at 37°C for 35 min and were purified with a MinElute PCR Purification Kit (QIAGEN). After the optimization of PCR cycle number using SYBER Green I Nucleic Acid gel Stain (Takara Bio), transposed fragments were amplified using NEBNext High Fidelity 2×PCR Master mix and index primers, and were purified with a MinElute PCR Purification Kit (QIAGEN). Library DNA was sized selected (240–360 bps) with BluePippin (Sage Science). Sequencing was performed using HiSeq1500 or HiSeq2500 (Illumina) with a single-read sequencing length of 60 bp. Bowtie2 (version 2.2.6; with default parameters) was used to map reads to the reference genome (UCSC/mm10) with annotation data from iGenomes (Illumina). Reads mapped to mitochondria were removed. To ensure even processing, reads were randomly downsampled from each sample to adjust to the smallest read number of samples. MACS (version 2.1.1; with default parameters) was used to call peaks in downsampled reads. The catalogue of all peaks called in any samples was produced by merging all called peaks that overlapped by at least one base pair using bedtools. The MACS bdgcmp function was used to compute the fold enrichment over the background for all populations, and the bedtools map function was used to count fragments in the catalogue in each population. Fragment counts at each site in the catalogue were quantile normalized between samples using the PreprocessCore package in R (3.3.2). We used the Homer package with command annotatePeaks.pl using default parameters to annotate regions with promoter and distal labels and the nearest gene, and with command findMotifsGenome.pl using default parameters to identify enriched motifs, and the catalogue of all called peaks as a background.

## Quantification and statistical analysis

Statistical tests were performed using Prism version 9 (GraphPad). The significance of difference was measured by the Student's *t*-test or one-way ANOVA. Data are shown as the mean ± SEM. Significance was taken at values of *$p < 0.05$, **$p < 0.01$, and ***$p < 0.001$.

## Acknowledgements

We thank Dr. Daniel G Tenen for providing us with Cebpa mutant mice. The super-computing resource was provided by The Human Genome Center, The Institute of Medical Science, The University of Tokyo. This work was supported in part by Grants-in-Aid for Scientific Research (#19H05653, #20K08728)

and Scientific Research on Innovative Area 'Replication of Non-Genomic Codes' (#19H05746) from Japanese Society for the Promotion of Science, and Moonshot (#21zf0127003h0001) and UTOPIA (JP223fa627001) projects from the Japan Agency for Medical Research and Development.

## Additional information

### Funding

| Funder | Grant reference number | Author |
|---|---|---|
| Japan Society for the Promotion of Science | 19H05653 | Atsushi Iwama |
| Japan Society for the Promotion of Science | 20K08728 | Yaeko Nakajima-Takagi |
| Japan Society for the Promotion of Science | 19H05746 | Atsushi Iwama |
| Japan Agency for Medical Research and Development | 21zf0127003h0001 | Atsushi Iwama |
| Japan Agency for Medical Research and Development | JP223fa627001 | Atsushi Iwama |

The funders had no role in study design, data collection and interpretation, or the decision to submit the work for publication.

### Author contributions

Yaeko Nakajima-Takagi, Conceptualization, Data curation, Formal analysis, Validation, Investigation, Writing - original draft; Motohiko Oshima, Data curation, Formal analysis; Junichiro Takano, Shuhei Koide, Naoki Itokawa, Shun Uemura, Shohei Andoh, Kazumasa Aoyama, Yusuke Isshiki, Daisuke Shinoda, Atsunori Saraya, Fumio Arai, Kiyoshi Yamaguchi, Yoichi Furukawa, Formal analysis, Investigation; Masayuki Yamashita, Writing – review and editing; Haruhiko Koseki, Resources; Tomokatsu Ikawa, Formal analysis, Investigation, Writing – review and editing; Atsushi Iwama, Conceptualization, Supervision, Funding acquisition, Writing – review and editing

### Author ORCIDs

Yaeko Nakajima-Takagi http://orcid.org/0000-0002-2341-2409
Shun Uemura http://orcid.org/0000-0002-1520-6808
Masayuki Yamashita http://orcid.org/0000-0001-9459-4329
Yoichi Furukawa http://orcid.org/0000-0003-0462-8631
Haruhiko Koseki http://orcid.org/0000-0001-8424-5854
Atsushi Iwama http://orcid.org/0000-0001-9410-8992

### Ethics

All experiments using mice were performed in accordance with our institutional guidelines for the use of laboratory animals and approved by the Review Board for Animal Experiments of Chiba University (approval ID: 30-56) and the University of Tokyo (approval ID: PA18-03).

### Decision letter and Author response

Decision letter https://doi.org/10.7554/eLife.83004.sa1
Author response https://doi.org/10.7554/eLife.83004.sa2

## Additional files

### Supplementary files
• MDAR checklist
• Supplementary file 1. List of gene sets for GSEA.

• Supplementary file 2. List of target genes of PRC1, PRC2, and PCGF1 and bivalent genes in LSK cells.

• Supplementary file 3. DNA motifs identified in ATAC peaks at proximal promoters enriched in *Pcgf1*$^{\Delta/\Delta}$ CD135$^-$ LSK cells (Top 12 motifs).

• Supplementary file 4. List of differentially expressed genes (DEGs) in GMPs and those overlapped between "5-FU day 8 vs day 0 GMPs" and "Control vs *Pcgf1*$^{\Delta/\Delta}$ GMPs.

• Supplementary file 5. List of oligonucleptide and TaqMan gene expression assay materials.

## Data availability

RNA sequence, ChIP sequence and ATAC sequence data were deposited in the DDBJ (accession numbers DRA008518 and DRA013523).

The following datasets were generated:

| Author(s) | Year | Dataset title | Dataset URL | Database and Identifier |
| --- | --- | --- | --- | --- |
| Iwama A | 2022 | Polycomb repressive complex 1.1 orchestrates homeostatic and emergency hematopoiesis and restricts transformation by acting as a rheostat of myelopoiesis | https://ddbj.nig. ac.jp/resource/ sra-submission/ DRA013523 | DDBJ, DRA013523 |
| Iwama A | 2019 | Non-canonical PRC1 orchestrates homeostatic and emergency hematopoiesis and restricts transformation by acting as a rheostat of myeloid differentiation | https://ddbj.nig. ac.jp/resource/ sra-submission/ DRA008518 | DDBJ, DRA008518 |

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

# Appendix 1

## Appendix 1—key resources table

| Reagent type (species) or resource | Designation | Source or reference | Identifiers | Additional information |
|---|---|---|---|---|
| Strain, strain background (*Mus musculus*) | Wild-type mice: C57BL/6 (CD45.2) | Japan SLC | C57BL6JJmsSlc | |
| Strain, strain background (*M. musculus*) | Wild-type mice: C57BL/6 (CD45.1) | Sankyo-Lab Service | | |
| Strain, strain background (*M. musculus*) | *Rosa26::Cre-ERT2*: C57BL/6-Gt(ROSA)26Sor^tm9(Cre/ESR1)Arte^ | TaconicArtemis GmbH | Model 10471 | |
| Strain, strain background (*M. musculus*) | *Pcgf1^fl^* | This paper | | Koseki H Lab |
| Strain, strain background (*M. musculus*) | *Cebpa^fl^*: B6.129S6(CBA)-*Cebpa^tm1Dgt/J^* | The Jackson Laboratory | JAX: 006447 | Daniel G. Tenen Lab |
| Cell line (*Homo sapiens*) | 293gpg | *Ory et al., 1996*. | | |
| Cell line (*M. musculus*) | TSt-4 | *Masuda et al., 2005*. | | |
| Cell line (*M. musculus*) | TSt-4/DLL1 | *Masuda et al., 2005*. | | |
| Antibody | Biotin anti-mouse Ly-6G/Ly-6C (Gr-1) (RB6-8C5) (rat monoclonal) | Tonbo Biosciences | Cat# 30-5931; RRID:AB_2621652 | FACS ($3.5 \times 10^{-3}$ µL/1 $\times 10^6$ cell) |
| Antibody | Biotin anti-human/mouse CD11b (M1/70) (rat monoclonal) | Tonbo Biosciences | Cat# 30-0112; RRID:AB_2621639 | FACS ($1.8 \times 10^{-2}$ µL/1 $\times 10^6$ cell) |
| Antibody | Biotin anti-mouse TER-119 (TER-119) (rat monoclonal) | Tonbo Biosciences | Cat# 30-5921; RRID:AB_2621651 | FACS ($3.5 \times 10^{-3}$ µL/1 $\times 10^6$ cell) |
| Antibody | Biotin anti-human/mouse CD45R (B220) (RA3-6B2) (rat monoclonal) | Tonbo Biosciences | Cat# 30-0452; RRID:AB_2621644 | FACS ($1.8 \times 10^{-2}$ µL/1 $\times 10^6$ cell) |
| Antibody | Biotin anti-mouse CD127 (IL-7Rα) antibody (rat monoclonal) | BioLegend | Cat# 121104; RRID:AB_312989 | FACS ($1.8 \times 10^{-2}$ µL/1 $\times 10^6$ cell) |
| Antibody | Biotin anti-mouse CD4 antibody (rat monoclonal) | BioLegend | Cat# 100404; RRID:AB_493502 | FACS ($9 \times 10^{-3}$ µL/1 $\times 10^6$ cell) |
| Antibody | Biotin anti-mouse CD8a (53–6.7) (rat monoclonal) | Tonbo Biosciences | Cat# 30-0081; RRID:AB_2621638 | FACS ($9 \times 10^{-3}$ µL/1 $\times 10^6$ cell) |
| Antibody | Biotin anti-mouse Ly-6A/E (Sca-1) antibody (rat monoclonal) | BioLegend | Cat# 108104; RRID:AB_3133418 | FACS ($2.5 \times 10^{-2}$ µL/1 $\times 10^6$ cell) |
| Antibody | CD41a monoclonal antibody (eBioMWReg30 (MWReg30)), Biotin (rat monoclonal) | eBioscience | Cat# 13-0411-85; RRID:AB_76348 | FACS ($0.1$ µL/1 $\times 10^6$ cell) |
| Antibody | PerCP/Cyanine5.5 Streptavidin | BioLegend | Cat# 405214; RRID:AB_2716577 | FACS ($5 \times 10^{-2}$ µL/1 $\times 10^6$ cell) |
| Antibody | APC anti-human/mouse CD45R (B220) (RA3-6B2) (rat monoclonal) | Tonbo Biosciences | Cat# 20-0452; RRID:AB_2621574 | FACS ($2.5 \times 10^{-2}$ µL/1 $\times 10^6$ cell) |
| Antibody | APC anti-mouse CD117 (c-Kit) antibody (rat monoclonal) | BioLegend | Cat# 105812; RRID:AB_313221 | FACS ($2.5 \times 10^{-2}$ µL/1 $\times 10^6$ cell) |
| Antibody | APC anti-human/mouse CD11b (M1/70) (rat monoclonal) | Tonbo Biosciences | Cat# 20-0112; RRID:AB_2621556 | FACS ($5 \times 10^{-3}$ µL/1 $\times 10^6$ cell) |
| Antibody | APC anti-mouse CD4 antibody (rat monoclonal) | BioLegend | Cat# 100412; RRID:AB_312697 | FACS ($0.1$ µL/1 $\times 10^6$ cell) |
| Antibody | APC/Cyanine7 anti-mouse CD4 antibody (rat monoclonal) | BioLegend | Cat# 100526; RRID:AB_312727 | FACS ($5 \times 10^{-2}$ µL/1 $\times 10^6$ cell) |
| Antibody | APC/Cyanine7 anti-mouse CD8a antibody (rat monoclonal) | BioLegend | Cat# 100714; RRID:AB_312753 | FACS ($5 \times 10^{-2}$ µL/1 $\times 10^6$ cell) |
| Antibody | APC/Cyanine7 anti-mouse CD48 antibody (Armenian hamster monoclonal) | BioLegend | Cat# 103431; RRID:AB_2621462 | FACS ($2.5 \times 10^{-2}$ µL/1 $\times 10^6$ cell) |
| Antibody | APC/Cyanine7 anti-mouse CD45.2 antibody (mouse monoclonal) | BioLegend | Cat# 109824; RRID:AB_830789 | FACS ($4 \times 10^{-2}$ µL/1 $\times 10^6$ cell) |

*Appendix 1 Continued on next page*

*Appendix 1 Continued*

| Reagent type (species) or resource | Designation | Source or reference | Identifiers | Additional information |
|---|---|---|---|---|
| Antibody | APC/Cyanine7 Streptavidin | BioLegend | Cat# 405208 | FACS ($5 \times 10^{-2}$ µL/1 $\times$ $10^6$ cell) |
| Antibody | PE anti-mouse Ly-6A/E (Sca-1) antibody (rat monoclonal) | BioLegend | Cat# 108108; RRID:AB_313345 | FACS ($2.5 \times 10^{-2}$ µL/1 $\times$ $10^6$ cell) |
| Antibody | PE anti-mouse/human CD11b antibody (rat monoclonal) | BioLegend | Cat# 101208; RRID:AB_312791 | FACS ($5 \times 10^{-3}$ µL/1 $\times$ $10^6$ cell) |
| Antibody | PE anti-mouse CD150 (SLAM) antibody (rat monoclonal) | BioLegend | Cat# 115904; RRID:AB_313683 | FACS (0.1 µL/1 $\times$ $10^6$ cell) |
| Antibody | PE anti-mouse CD16/32 antibody (rat monoclonal) | BioLegend | Cat# 101308; RRID:AB_312807 | FACS ($2.5 \times 10^{-2}$ µL/1 $\times$ $10^6$ cell) |
| Antibody | PE anti-mouse CD127 (IL-7Rα) antibody (rat monoclonal) | BioLegend | Cat# 135010; RRID:AB_1937251 | FACS (0.1 µL/1 $\times$ $10^6$ cell) |
| Antibody | PE anti-mouse Ly-6G/Ly-6C (Gr-1) antibody (rat monoclonal) | BioLegend | Cat# 108408; RRID:AB_313373 | FACS ($2.5 \times 10^{-3}$ µL/1 $\times$ $10^6$ cell) |
| Antibody | PE anti-mouse CD8a antibody (rat monoclonal) | BioLegend | Cat# 100708; RRID:AB_312747 | FACS (0.2 µL/1 $\times$ $10^6$ cell) |
| Antibody | PE anti-mouse CD43 antibody (rat monoclonal) | BioLegend | Cat# 143206; RRID:AB_11124719 | FACS ($2.5 \times 10^{-2}$ µL/1 $\times$ $10^6$ cell) |
| Antibody | PE anti-mouse F4/80 antibody (rat monoclonal) | BioLegend | Cat# 123110; RRID:AB_8934865 | FACS (0.1 µL/1 $\times$ $10^6$ cell) |
| Antibody | PE/Cyanine7 anti-mouse Ly-6A/E (Sca-1) antibody (rat monoclonal) | BioLegend | Cat# 108114; RRID:AB_493596 | FACS ($2.5 \times 10^{-2}$ µL/1 $\times$ $10^6$ cell) |
| Antibody | PE/Cyanine7 anti-mouse CD45.1 antibody (mouse monoclonal) | BioLegend | Cat# 110730; RRID:AB_1134168 | FACS ($4 \times 10^{-2}$ µL/1 $\times$ $10^6$ cell) |
| Antibody | CD34 monoclonal antibody (RAM34), FITC (rat monoclonal) | eBioscience | Cat# 11-0341-85; RRID:AB_1465022 | FACS (0.25 µL/1 $\times$ $10^6$ cell) |
| Antibody | FITC anti-mouse Ly-6D antibody (rat monoclonal) | BioLegend | Cat# 138606; RRID:AB_11203888 | FACS ($2.5 \times 10^{-2}$ µL/1 $\times$ $10^6$ cell) |
| Antibody | FITC anti-mouse IgM antibody (rat monoclonal) | BioLegend | Cat# 406506; RRID:AB_315056 | FACS ($5 \times 10^{-2}$ µL/1 $\times$ $10^6$ cell) |
| Antibody | FITC anti-mouse Ly6C antibody (rat monoclonal) | BioLegend | Cat# 128005; RRID:AB_1186134 | FACS (0.1 µL/1 $\times$ $10^6$ cell) |
| Antibody | Ki-67 monoclonal antibody (SolA15), FITC (rat monoclonal) | eBioscience | Cat# 11-5698-82; RRID:AB_11151330 | FACS (0.25 µL/1 $\times$ $10^6$ cell) |
| Antibody | Alexa Fluor 488 anti-mouse CD105 antibody (rat monoclonal) | BioLegend | Cat# 120406; RRID:AB_961053 | FACS (0.25 µL/1 $\times$ $10^6$ cell) |
| Antibody | violetFluor 450 anti-mouse CD45.2 (104) (mouse monoclonal) | Tonbo Biosciences | Cat# 75-0454; RRID:AB_2621950 | FACS ($4 \times 10^{-2}$ µL/1 $\times$ $10^6$ cell) |
| Antibody | Brilliant Violet 421 anti-mouse CD135 antibody (rat monoclonal) | BioLegend | Cat# 135314; RRID:AB_2562339 | FACS (0.25 µL/1 $\times$ $10^6$ cell) |
| Antibody | Brilliant Violet 421 anti-mouse CD150 (SLAM) antibody (rat monoclonal) | BioLegend | Cat# 115925; RRID:AB_10896787 | FACS (0.1 µL/1 $\times$ $10^6$ cell) |
| Antibody | Brilliant Violet 421 anti-mouse Ly6G antibody (rat monoclonal) | BioLegend | Cat# 127627; RRID:AB_10897944 | FACS (0.1 µL/1 $\times$ $10^6$ cell) |
| Antibody | Brilliant Violet 510 anti-mouse CD16/32 antibody (rat monoclonal) | BioLegend | Cat# 101333; RRID:AB_2563692 | FACS ($2.5 \times 10^{-2}$ µL/1 $\times$ $10^6$ cell) |
| Antibody | CD34 monoclonal antibody (RAM34), eFluor 450 (rat monoclonal) | eBioscience | Cat# 48-0341-82; RRID:AB_2043837 | FACS (0.25 µL/1 $\times$ $10^6$ cell) |
| Antibody | Brilliant Violet 605 Streptavidin | BioLegend | Cat# 405229 | FACS ($5 \times 10^{-2}$ µL/1 $\times$ $10^6$ cell) |
| Antibody | V500 mouse anti-mouse CD45.2 (mouse monoclonal) | BD Biosciences | Cat# 562129; RRID:AB_10897142 | FACS ($4 \times 10^{-2}$ µL/1 $\times$ $10^6$ cell) |

*Appendix 1 Continued on next page*

*Appendix 1 Continued*

| Reagent type (species) or resource | Designation | Source or reference | Identifiers | Additional information |
|---|---|---|---|---|
| Antibody | BV510 Mouse Anti-Mouse CD45.1 (mouse monoclonal) | BD Biosciences | Cat# 565278; RRID:AB_2739150 | FACS ($4 \times 10^{-2}$ µL/$1 \times 10^6$ cell) |
| Antibody | Ubiquityl-Histone H2A (Lys119) (D27C4) XP rabbit mAb (rabbit monoclonal) | Cell Signaling Technology | Cat# 8240S; RRID:AB_10891618 | ChIP (2 µg) |
| Antibody | Anti-trimethyl-Histone H3 (Lys27) antibody (rabbit polyclonal) | Millipore | Cat# 07-449; RRID:AB_310624 | ChIP (2 µg) |
| Antibody | Monoclonal anti-FLAG M2, antibody produced in mouse (mouse monoclonal) | Sigma | Cat# F1084 | ChIP (2 µg) |
| Antibody | β-Catenin (6B3) rabbit mAb (rabbit monoclonal) | Cell Signaling Technology | Cat# 9582S; RRID:AB_823447 | IF (1:100) |
| Antibody | Human/mouse CD117/c-kit antibody (goat polyclonal) | R&D Systems | Cat# AF1356; RRID:AB_354750 | IF (1:100) |
| Antibody | CD41a monoclonal antibody (eBioMWReg30 (MWReg30)), Biotin (rat monoclonal) | eBioscience | Cat# 13-0411-85; RRID:AB_763489 | IF (1:200) |
| Antibody | Alexa Fluor 647 anti-mouse CD16/32 antibody (rat monoclonal) | BioLegend | Cat# 101314; RRID:AB_2278396 | IF (1:50) |
| Antibody | Biotin anti-mouse Ly-6A/E (Sca-1) antibody (rat monoclonal) | BioLegend | Cat# 108104; RRID:AB_313341 | IF (1:100) |
| Antibody | Streptavidin, Alexa Fluor 488 conjugate | Invitrogen | Cat# S11223 | IF (1:200) |
| Antibody | Donkey anti-rabbit IgG (H+L), highly cross-adsorbed antibody, CF488A conjugated (donkey polyclonal) | Biotium | Cat# 20015; RRID:AB_10559669 | IF (1:400) |
| Antibody | Donkey anti-goat IgG (H+L) cross-adsorbed secondary antibody, Alexa Fluor 555 (donkey polyclonal) | Invitrogen | Cat# A-21432; RRID:AB_2535853 | IF (1:400) |
| Antibody | Donkey anti-goat IgG (H+L) cross-adsorbed secondary antibody, Alexa Fluor 568 (donkey polyclonal) | Invitrogen | Cat# A-11057; RRID:AB_2534104 | IF (1:400) |
| Sequence-based reagent | Primer pairs for genotyping | This study | | *Supplementary file 5* |
| Sequence-based reagent | Primer pairs for qPCR | Universal ProbeLibrary Assay Design Center (Roche) | | *Supplementary file 5* |
| Sequence-based reagent | Primer pairs for ChIP-qPCR | This study | | *Supplementary file 5* |
| Peptide, recombinant protein | Micrococcal Nuclease | New England BioLabs | M0247S | |
| Peptide, recombinant protein | RNase A | Sigma-Aldrich | R6513 | |
| Peptide, recombinant protein | Proteinase K | Roche | 3115852001 | |
| Peptide, recombinant protein | Recombinant mouse IL-7 (carrier-free) | BioLegend | 577802 | |
| Peptide, recombinant protein | Recombinant mouse SCF (carrier-free) | BioLegend | 579706 | |
| Peptide, recombinant protein | Recombinant human TPO (carrier-free) | BioLegend | 763706 | |
| Peptide, recombinant protein | Recombinant mouse IL-3 (carrier-free) | BioLegend | 575506 | |
| Peptide, recombinant protein | Recombinant murine GM-CSF | PeproTech | 315-03 | |
| Peptide, recombinant protein | Recombinant human Flt3-ligand | PeproTech | 300-19 | |

*Appendix 1 Continued on next page*

*Appendix 1 Continued*

| Reagent type (species) or resource | Designation | Source or reference | Identifiers | Additional information |
|---|---|---|---|---|
| Peptide, recombinant protein | Recombinant murine IL-11 | PeproTech | 220-11 | |
| Commercial assay or kit | RNeasy Micro Plus Kit | QIAGEN | 74034 | |
| Commercial assay or kit | TRI Reagent LS | MOR | TS120 | |
| Commercial assay or kit | SuperScript IV First-Strand Synthesis System | Invitrogen | 18091050 | |
| Commercial assay or kit | ReverTra Ace a- | TOYOBO | FSQ-301 | |
| Commercial assay or kit | FastStart Universal Probe Master | Roche | 4914058001 | |
| Commercial assay or kit | Methocult M3234 | STEMCELL Technologies | 03234 | |
| Commercial assay or kit | BD Phosflow Lyse/Fix Buffer | BD Biosciences | 558049 | |
| Commercial assay or kit | BD Phosflow Perm Buffer II | BD Biosciences | 558052 | |
| Commercial assay or kit | TB Green Premix Ex Taq II | Takara Bio | RR820S | |
| Commercial assay or kit | cOmplete, Mini, Protease Inhibitor Cocktail | Roche | 11836170001 | |
| Commercial assay or kit | Dynabeads M-280 sheep anti-rabbit IgG | Invitrogen | 11203D | |
| Commercial assay or kit | Dynabeads M-280 sheep anti-mouse IgG | Invitrogen | 11202D | |
| Commercial assay or kit | MinElute PCR Purification Kit | QIAGEN | 28006 | |
| Commercial assay or kit | SMART-Seq v4 Ultra Low Input RNA Kit | Clontech | 634890 | |
| Commercial assay or kit | NEBNext Ultra DNA Library Prep Kit | New England BioLabs | E7370 | |
| Commercial assay or kit | ThruPLEX DNA-seq Kit | Clontech | R400428 | |
| Commercial assay or kit | Tagment DNA buffer | Illumina | 15027866 | |
| Commercial assay or kit | Tagment DNA enzyme | Illumina | 15027865 | |
| Commercial assay or kit | SYBER Green I Nucleic Acid gel Stain | Takara Bio | 5761A | |
| Commercial assay or kit | NEBNext High Fidelity 2x PCR Master mix | New England BioLabs | M0541S | |
| Commercial assay or kit | Chromium Next GEM Single Cell 3' Reagent Kits v3.1 | 10X Genomics | 1000269 | |
| Commercial assay or kit | O.T.C Compound | Sakura | D3571 | |
| Commercial assay or kit | ProLong Glass Antifade Mountant | Thermo Scientific | P36982 | |
| Commercial assay or kit | Avidin/Biotin Blocking Kit | VECTOR | SP-2001 | |
| Chemical compound, drug | Tamoxifen | Sigma-Aldrich | T5648 | |
| Chemical compound, drug | 4-Hydroxytamoxifen | Sigma-Aldrich | H7904 | |
| Chemical compound, drug | Corn oil | Sigma-Aldrich | C8267 | |
| Chemical compound, drug | Propidium iodide | Sigma-Aldrich | P4170 | |
| Chemical compound, drug | 7-AAD | Sigma-Aldrich | A9400 | 1 µg/mL |
| Chemical compound, drug | 2-Mercaptoethanol (2-ME) | Sigma-Aldrich | M3148 | |
| Chemical compound, drug | Formaldehyde solution (FA) | Sigma-Aldrich | F8775 | |
| Chemical compound, drug | IGEPAL CA-630 | Sigma-Aldrich | I8896 | |
| Chemical compound, drug | Sodium pyrubate | Thermo Scientific | 11360070 | |
| Chemical compound, drug | DAPI | Invitrogen | D3571 | 1 µg/mL |

*Appendix 1 Continued on next page*

*Appendix 1 Continued*

| Reagent type (species) or resource | Designation | Source or reference | Identifiers | Additional information |
|---|---|---|---|---|
| Chemical compound, drug | Pierce 16% formaldehyde (w/v), methanol-free | Thermo Scientific | 28906 | 8% |
| Chemical compound, drug | 5-fluorouracil (5-FU) | Kyowa KIRIN | 5-FU 250 mg | 3.75 mg per 25 g body weight mouse |
| Software, algorithm | Prism 9 | GraphPad | | https://www.graphpad.com/ |
| software, algorithm | L-Calc | STEMCELL Technologies | | https://www.stemcell.com/l-calc-software.html |
| Software, algorithm | Gene set enrichment analysis | Broad Institute | | http://software.broadinstitute.org/gsea/index.jsp |
| Software, algorithm | g:Profiler | Raudvere et al., 2019. | | https://biit.cs.ut.ee/gprofiler/ |
| Software, algorithm | Integrate Genomics Viewer v2.4.4 | Broad Institute | | https://software.broadinstitute.org/software/igv/ |
| Software, algorithm | Fiji | *Schindelin et al., 2012*. | | https://fiji.sc |
| Software, algorithm | ImageJ | National Institutes of Health | | https://imagej.nih.gov/ij/index.html |
| Other | Normal donkey serum | Jackson ImmunoResearch | 017-000-121 | Blocking reagent for immunostaining |
| Other | S-Clone SF-O3 | EIDIA Co., Ltd | 1303 | Medium for cell cluture |
| Other | 10% bovine serum albumin in Iscove's MDM | STEMCELL Technologies | 09300 | Supplement for cell cluture |

