## [Editor Report]

The authors present a manuscript aiming to understand the mechanism(s) underlying myeloid bias in HSCs, specifically focused on the role of Pcgf1, and therefore PRC1.1, in the regulation of hematopoiesis. The current study demonstrates that conditional deletion of Pcgf1 in HSCs causes a lineage switch from lymphoid to myeloid fates and that a key mechanism for this lineage switch is regulation of the H2AK119ub1 chromatin mark, leading to de-repression of CEBPalpha, a key transcription factor that promotes myeloid cell fate. This important work is of interest to the community of researchers interested in myeloid differentiation, lineage fate decisions in hematopoietic stem cells, and the molecular mechanisms that contribute to the initiation of myeloid malignancies. The methods are rigorous and the results convincingly support the authors' conclusions.

---

## [Decision Letter]

**Decision letter after peer review:**

Thank you for submitting your article "Polycomb repressive complex 1.1 coordinates homeostatic and emergency myelopoiesis" for consideration by *eLife*. Your article has been reviewed by 3 peer reviewers, and the evaluation has been overseen by a Reviewing Editor and Mone Zaidi as the Senior Editor. The following individuals involved in the review of your submission have agreed to reveal their identity: Carl R Walkley (Reviewer #2).

The authors present a manuscript aiming to understand the mechanism(s) underlying myeloid bias in HSCs, specifically focused on the role of Pcgf1, and therefore PRC1.1, in the regulation of hematopoiesis. This work is of interest to the community of researchers interested in myeloid differentiation, lineage fate decisions in HSCs, and the molecular mechanisms that contribute to the initiation of myeloid malignancies. This work also is of novel significance for translational purposes and could lead to the design of better therapeutics for myeloid malignancies, in the least the utility of the novel mouse model presented in this work. The current study demonstrates that conditional deletion of Pcgf1 in HSCs causes a lineage switch from lymphoid to myeloid fates and that a key mechanism for this lineage switch is regulation of the H2AK119ub1 chromatin mark, leading to de-repression of CEBPalpha, a key transcription factor that promotes myeloid cell fate. The authors also demonstrate that deletion of Pcgf1 in HSCs can also lead to deregulated myelopoiesis, leading to a lethal MPN in a subset of animals, implicating Pcgf1 as a critical regulator of emergency myelopoiesis with the potential to prevent malignant transformation of myeloid progenitors.

The reviewers found significant merit in the work both technically and conceptually. However, the evaluation identified weaknesses that prevent the publication of this work in the present form. The main shortcomings of the presented work include:

1) Lack of clarity on whether the KO findings result in a normal but excessive myeloid differentiation, or from a myeloid skew with a block in the maturation of the mature myeloid cells;

2) Need for further delineation of the subtypes of mature myeloid cells in vivo;

3) Provide in vivo data to support the in vitro evidence that restoring CEBPa rescues the myeloid bias in the Pcgf1 KO HSPCs.

A complete list of reviewer comments can be found below.

*Reviewer #1 (Recommendations for the authors):*

1. The authors show that HoxA9 overexpression is associated with the activation of b-catenin, but the mechanism for this is unclear. Is HoxA9 a direct transcriptional activator of b-catenin? If not, can the authors speculate about a mechanism for the increase in β-catenin activation?

2. Is this really "emergency myelopoiesis", or is it pathologic myelopoiesis? The authors state that "the production of circulating myeloid cells was not enhanced". Is that because there is also a differentiation block? Addressing the question in of whether there is an expansion of undifferentiated cells would help to make this distinction. In Figure 6, if there is an increase in self-renewing GMPs in Pcgf1 KO mice with increased HoxA9 expression, why do the mice develop MPN and not AML? Is there any differentiation block along with the myeloid expansion? What happens to c-kit expression? Is this disease transplantable? It would also be helpful to see some high-power cytospins of the bone marrow or spleens of mice with MPN, to better assess the differentiation status.

3. It is suggested that Figure S6H should be in the last main figure, as it is a really helpful summary of the roles of PRC 1.1 in hematopoiesis.

*Reviewer #2 (Recommendations for the authors):*

Specific comments:

1. The work clearly shows a myeloid bias in hematopoiesis. It is not clear from the data as presented whether this is resulting in a normal but excessive myeloid differentiation, or from a myeloid skew with a block in the maturation of the mature myeloid cells. This is an important difference and would be important to understand. The present data demonstrate that the mice accumulate GMP and myeloid progenitors, but they also develop MPN and myeloid malignant/pre-malignant states. An alternative interpretation could be that the mature myeloid cells are not differentiating normally and that this is leading to excessive myeloid production to try to generate a functionally competent myeloid population for host defence.

2. The authors should more clearly describe the mature myeloid cells in the animals (across PB, BM, Spleen) and it would be useful to provide FACS plots of Mac-1/Gr-1 profiles (+/- F4/80, Ly6C/G).

3. The authors describe myeloid cells in the figure legends as "Mac-1+ and/or Gr-1+ myeloid cells" positive cells – please be more specific and define the populations. This applies throughout the manuscript and assays.

4. In the data in Figure 2E, can the authors provide qPCR or similar levels of Cebpa in the different genotypes. Is it demonstrated that germ-line heterozygous Cebpa would cause a reduction in the expression of Cebpa?

5. Did the authors age any mice post-5-FU to determine if this was sufficient to predispose to MPN generation or induce a more pronounced block in myeloid maturation?

6. The Hoxa9 reexpression studies in Figure 5G – This doesn't appear to have a WT control. The overexpression of Hoxa9 may be expected to have the same effect in a WT cell as that displayed (I am assuming this is a Pcgf1 deficient cell but this isn't completely clear to me from the text or legend).

7. Include an n for sick mice in Figure 6 (25 in total aged but how many got what diagnosis). Related to this, but potentially beyond the scope, is the disease transplantable?

*Reviewer #3 (Recommendations for the authors):*

Line 127: KO mice have mild anemia and leukopenia. Figure S2A is the only data associated with this statement – please show neutrophil, monocyte, and lymphocyte counts separately.

Figure 1I: Green/Red bar plotting could be problematic for people with color blindness. The labeling of the HSPC populations underneath the genotypes is poorly aligned – which adds to the confusion. Because the bars are stacked with multiple colors, the statistical significance is indicated for the total or specific colored portion?

How does PCGF1 non-cell autonomously control hematopoiesis? What % of Pcgf1Δ/Δ cells is needed to achieve this non-cell autonomous inhibition? This seems to be a potent aspect of regulation – some discussion would be helpful.

Line 167 and 169, "… cells display slower growth under HSPC expanding condition" and "better growth". The word "growth" in these sentences is more accurate to be replaced by "population doubling".

S5A is never referenced in the text. S5B, middle plot. Significance is indicated by a star in the B cell portion of the stacked bars. This is for comparing which samples/cell types, as multiple ways of comparison can be made.

Figure 2D/E provided in vitro results for restoring CEBPa in rescuing the myeloid bias in the Pcgf1Δ/Δ HSPCs. Support this claim with in vivo results.

Line 242 refers to Figure S3D: there is no panel D in Figure S3.

Line 311 Figure 3D – this should have been 3G, if I understood correctly. If I did not understand correctly, then Figure 3G is never cited. This is so terribly confusing!

Figure 3I – so poorly aligned/labelled.

Line 320-322 "Moreover, ATAC-seq analyses also revealed trend toward enrichment of ATAC peaks containing CEBP motif in Pcgf1Δ/Δ HSPCs compared to the control, suggesting that chromatin regions containing CEBP motif tended to open in Pcgf1Δ/Δ HSPCs (Figure 3I)." The figure supporting this claim seems to have "p=0.3". Such a p-value is "ns" everywhere else. Why is it different here? As far as this reviewer can gather, the only significant difference in this figure is between control LSK and control GMP. Therefore, these results cannot and should not be cited as evidence to support "chromatin regions containing CEBP motif tended to open in HSPCs (Figure 3I)." Authors should remove any misleading claim based on this strenuous comparison.

Figure 4A. Can the difference in cluster GMPs be quantified?

Line 412-413 figure legend for Figure 4C: "Each symbol is derived from an individual culture". This is supposed to be from in vivo experiments? No statistical significance indicated?

Figure 5B,5G/H, states that HSC cultures produced GMPs. This statement is misleading and contributes to the over-interpretation of the results. To the best that this data can support, perhaps it is fair to say cells expressing immunophenotypic GMP markers are produced. The authors interpreted the difference in myeloid culture condition 1 and 2 being the different responses of HSC and GMP toward the loss of Pcgf1. However, it is equally possible that the different growth factors (type and concentrations) caused the differential response between the control and Pcgf1Δ/Δ cells. The authors need to culture HSC and GMP in both myeloid culture condition 1 and 2 to justify their claims.

Figure 5D. The images are missing the labels of genotypes.

Figure 5E legend says the samples are the GMPs from HSC cultures in 5B. Were these "GMPs" sorted? Please specify. Why are the day 16 control "GMPs" missing?

Figure 5F. What samples are being compared by RNA-seq? This reviewer is guessing the same samples shown in Figure 5E. If so, it begs the same question as to why are the day 16 control "GMPs" missing, and why are the day 0 Pcgf1Δ/Δ GMPs missing.

---

## [Author Response]

The authors present a manuscript aiming to understand the mechanism(s) underlying myeloid bias in HSCs, specifically focused on the role of Pcgf1, and therefore PRC1.1, in the regulation of hematopoiesis. This work is of interest to the community of researchers interested in myeloid differentiation, lineage fate decisions in HSCs, and the molecular mechanisms that contribute to the initiation of myeloid malignancies. This work also is of novel significance for translational purposes and could lead to the design of better therapeutics for myeloid malignancies, in the least the utility of the novel mouse model presented in this work. The current study demonstrates that conditional deletion of Pcgf1 in HSCs causes a lineage switch from lymphoid to myeloid fates and that a key mechanism for this lineage switch is regulation of the H2AK119ub1 chromatin mark, leading to de-repression of CEBPalpha, a key transcription factor that promotes myeloid cell fate. The authors also demonstrate that deletion of Pcgf1 in HSCs can also lead to deregulated myelopoiesis, leading to a lethal MPN in a subset of animals, implicating Pcgf1 as a critical regulator of emergency myelopoiesis with the potential to prevent malignant transformation of myeloid progenitors.The reviewers found significant merit in the work both technically and conceptually. However, the evaluation identified weaknesses that prevent the publication of this work in the present form. The main shortcomings of the presented work include:1) Lack of clarity on whether the KO findings result in a normal but excessive myeloid differentiation, or from a myeloid skew with a block in the maturation of the mature myeloid cells;2) Need for further delineation of the subtypes of mature myeloid cells in vivo;3) Provide in vivo data to support the in vitro evidence that restoring CEBPa rescues the myeloid bias in the Pcgf1 KO HSPCs.A complete list of reviewer comments can be found below.

Thank you for your suggestions. In response to the points raised above, we performed additional experiments, presented data in new figures, and revised the main text. Among the 3 main shortcomings of the presented work you pointed out, we clearly addressed the points 1 and 2. Regarding the point 3, we tried competitive BM transplantation assays using conditional *Pcgf1* KO, *Cebpa* KO, and compound mice to rescue dysregulated differentiation of *Pcgf1^Δ/Δ^* HSPCs. However, we found that the deletion efficiency of *Cebpa* was poor in vivo using tamoxifen, while it is quite efficient in vitro using 4-hydroxytamoxifen. The data we obtained was hard to interpret. We would like to address this issue in our future experiments using different approaches. Details are described in the following responses to the reviewers.

Reviewer #1 (Recommendations for the authors):1. The authors show that HoxA9 overexpression is associated with the activation of b-catenin, but the mechanism for this is unclear. Is HoxA9 a direct transcriptional activator of b-catenin? If not, can the authors speculate about a mechanism for the increase in β-catenin activation?

We examined the promoter region of the *Ctnnb1* gene by JASPER database (https://jaspar.genereg.net/) and found at least one predicted sequence for HoxA9 binding (CTTATAAATCG) (data not shown). We performed qPCR analysis using *HoxA9* overexpression samples at day 12 of culture, in which nuclear localization of β-catenin was observed upon *HoxA9* overexpression (Figure 5H). *Ctnnb1* gene expression was slightly but significantly upregulated in *HoxA9* overexpression samples compared to mock control (Figure 5—figure supplement 1C). These results raise the possibility that HoxA9 is a direct transcriptional activator of the *Ctnnb1* gene, but this warrants further analysis. We described these points in lines 385-392.

2. Is this really "emergency myelopoiesis", or is it pathologic myelopoiesis? The authors state that "the production of circulating myeloid cells was not enhanced". Is that because there is also a differentiation block? Addressing the question in of whether there is an expansion of undifferentiated cells would help to make this distinction. In Figure 6, if there is an increase in self-renewing GMPs in Pcgf1 KO mice with increased HoxA9 expression, why do the mice develop MPN and not AML? Is there any differentiation block along with the myeloid expansion? What happens to c-kit expression? Is this disease transplantable? It would also be helpful to see some high-power cytospins of the bone marrow or spleens of mice with MPN, to better assess the differentiation status.

We think the reviewer’s points are very important. The data in Figure S2 (Figure 1—figure supplement 2 in revised manuscript) indicate accumulation of mature myeloid cells in Pcgf1 KO BM and spleen. To further confirm this, we performed additional FACS analyses using mature myeloid surface marker (Ly6G, Ly6C, F4/80) antibodies. The results showed that the number of mature myeloid cells expressing myeloid differentiation markers increased in Pcgf1 KO BM and spleen. Monocytes but not neutrophils or F4/80+ macrophages significantly increased in Pcgf1 KO BM while both neutrophils and monocytes significantly increased in Pcgf1 KO spleen. We replaced the flow cytometric profiles of Mac1 and Gr1 staining with those of Mac1, Ly6G, and Ly6c staining and added graphs of their absolute numbers (Figure 1—figure supplement 2C and D). These findings indicate that the expanded Pcgf1 KO GMPs can supply mature myeloid cells and do not show evident differentiation block. This notion was supported by better production of Mac-1+ mature myeloid cells by Pcgf1 KO GMPs than control GMPs in culture (Figure 5—figure supplement 1A). Even in Pcgf1 KO mice at the MPN stage, GMPs actively produced mature myeloid cells and no obvious differentiation block like AML was observed. As we discussed, the molecular machineries that drive emergency myelopoiesis are often hijacked by transformed cells (Hérault et al., 2017). It is assumed that the epigenetic status that allows GMP self-renewal was enforced over time in the absence of Pcgf1, leading to enhanced production of mature myeloid cells that mimic MPN. Indeed, we did not see accumulation of c-Kit+ leukemic blasts in BM by FACS or accumulation of leukemic blasts in BM images with higher magnification (Figure 6F). Although we haven't investigated whether the disease is transplantable or not, we would like to address this in our future experiments. We described these points in lines 131-138 and 400-405.

3. It is suggested that Figure S6H should be in the last main figure, as it is a really helpful summary of the roles of PRC 1.1 in hematopoiesis.

Thank you for your suggestion. We presented Figure S6H as Figure 6H.

Reviewer #2 (Recommendations for the authors):Specific comments:1. The work clearly shows a myeloid bias in hematopoiesis. It is not clear from the data as presented whether this is resulting in a normal but excessive myeloid differentiation, or from a myeloid skew with a block in the maturation of the mature myeloid cells. This is an important difference and would be important to understand. The present data demonstrate that the mice accumulate GMP and myeloid progenitors, but they also develop MPN and myeloid malignant/pre-malignant states. An alternative interpretation could be that the mature myeloid cells are not differentiating normally and that this is leading to excessive myeloid production to try to generate a functionally competent myeloid population for host defence.

We think the reviewer’s points are very important. The data in Figure S2 (Figure 1—figure supplement 2 in revised manuscript) indicate accumulation of mature myeloid cells in Pcgf1 KO BM and spleen. To further confirm this, we performed additional FACS analyses using mature myeloid surface marker (Ly6G, Ly6C, F4/80) antibodies. The results showed that the number of mature myeloid cells expressing myeloid differentiation markers increased in Pcgf1 KO BM and spleen. Monocytes but not neutrophils or F4/80+ macrophages significantly increased in Pcgf1 KO BM while both neutrophils and monocytes significantly increased in Pcgf1 KO spleen. We replaced the flow cytometric profiles of Mac1 and Gr1 staining with those of Mac1, Ly6G, and Ly6c staining and added graphs of their absolute numbers (Figure 1—figure supplement 2C and D). These findings indicate that the expanded Pcgf1 KO GMPs can supply mature myeloid cells and do not show evident differentiation block. This notion was supported by better production of Mac-1+ mature myeloid cells by Pcgf1 KO GMPs than control GMPs in culture (Figure 5—figure supplement 1A). Even in Pcgf1 KO mice at the MPN stage, GMPs actively produced mature myeloid cells and no obvious differentiation block like AML was observed. As we discussed, the molecular machineries that drive emergency myelopoiesis are often hijacked by transformed cells (Hérault et al., 2017). It is assumed that the epigenetic status that allows GMP self-renewal was enforced over time in the absence of Pcgf1, leading to enhanced production of mature myeloid cells that mimic MPN. Indeed, we did not see accumulation of c-Kit+ leukemic blasts in BM by FACS or accumulation of leukemic blasts in BM images with higher magnification (Figure 6F). We described these points in lines 131-138 and 400- 405.

2. The authors should more clearly describe the mature myeloid cells in the animals (across PB, BM, Spleen) and it would be useful to provide FACS plots of Mac-1/Gr-1 profiles (+/- F4/80, Ly6C/G).

Thank you for your valuable suggestion. As described in response to your point 1, we performed an additional FACS analysis and presented the results in Figure 1—figure supplement 2C and D.

3. The authors describe myeloid cells in the figure legends as "Mac-1+ and/or Gr-1+ myeloid cells" positive cells – please be more specific and define the populations. This applies throughout the manuscript and assays.

For PB analysis, we used a mixture of Mac-1 and Gr-1 antibodies with the same conjugation. So, we defined myeloid cells as those positively reacted with a mixture of Mac-1 and Gr-1 antibodies. In this case, all types of myeloid cells were included. Instead, we performed detailed profiling of BM and spleen myeloid cells using Mac-1, Ly6C and Ly6G antibodies as described above (Figure 1—figure supplement 2C and D).

4. In the data in Figure 2E, can the authors provide qPCR or similar levels of Cebpa in the different genotypes. Is it demonstrated that germ-line heterozygous Cebpa would cause a reduction in the expression of Cebpa?

We have added qPCR data to Figure 2E. *Cebpa* gene expression levels in *Pcgf1^Δ/Δ^* ;*Cebpa^Δ/+^* cells were significantly reduced (Figure 2E lower right).

5. Did the authors age any mice post-5-FU to determine if this was sufficient to predispose to MPN generation or induce a more pronounced block in myeloid maturation?

Thank you for your very interesting suggestion. We would like to test in our future experiments.

6. The Hoxa9 reexpression studies in Figure 5G – This doesn't appear to have a WT control. The overexpression of Hoxa9 may be expected to have the same effect in a WT cell as that displayed (I am assuming this is a Pcgf1 deficient cell but this isn't completely clear to me from the text or legend).

We apologize for the lack of clarity in our description. The experiment in Figure 5G was performed using *Hoxa9*-overexpressing WT LSK cells, not in Pcgf1 KO cells. This experiment was designed to test whether *Hoxa9* overexpression alone can maintain the immunophenotypic GMPs for a long period as in Pcgf1 KO cells.

7. Include an n for sick mice in Figure 6 (25 in total aged but how many got what diagnosis). Related to this, but potentially beyond the scope, is the disease transplantable?

We have indicated the number of mice per diagnosis in Figure 6A. Although we haven't investigated whether the disease is transplantable or not, we would like to address this in our future experiments.

Reviewer #3 (Recommendations for the authors):Line 127: KO mice have mild anemia and leukopenia. Figure S2A is the only data associated with this statement – please show neutrophil, monocyte, and lymphocyte counts separately.

Thank you for your suggestion. We showed the lineage cell counts (myeloid/B/T) 1 and 6 months after tamoxifen injection in Figure 1—figure supplement 2B. We also analyzed myeloid cells (neutrophils, monocytes, and macrophages) more in detail in the BM and spleen and presented the data in Figure 1—figure supplement 2C and D. We described these points in lines 131-138.

Figure 1I: Green/Red bar plotting could be problematic for people with color blindness. The labeling of the HSPC populations underneath the genotypes is poorly aligned – which adds to the confusion. Because the bars are stacked with multiple colors, the statistical significance is indicated for the total or specific colored portion?

Thank you for your important point. We changed the bar plotting color from Green/Red to Blue/Red. We also improved the labeling of genotypes and cell populations to make it easier to understand. Statistical significance is based on the overall number of cells. We indicated this point in Figure 1I legend.

How does PCGF1 non-cell autonomously control hematopoiesis? What % of Pcgf1Δ/Δ cells is needed to achieve this non-cell autonomous inhibition? This seems to be a potent aspect of regulation – some discussion would be helpful.

Thank you very much for your very interesting suggestion. These findings suggest that *Pcgf1^Δ/Δ^* hematopoietic cells suppress expansion of both *Pcgf1^Δ/Δ^* and co-existing WT HSPCs through non- autonomous mechanisms. It is possible that accumulating *Pcgf1^Δ/Δ^* mature myeloid cells produce inflammatory cytokines that are suppressive to HSPCs. We are very interested in this issue and would like to address it in our future project. We discussed this point in lines 169-173.

Line 167 and 169, "… cells display slower growth under HSPC expanding condition" and "better growth". The word "growth" in these sentences is more accurate to be replaced by "population doubling".

As the reviewer pointed out, we replaced the word "growth" with "population doubling".

S5A is never referenced in the text. S5B, middle plot. Significance is indicated by a star in the B cell portion of the stacked bars. This is for comparing which samples/cell types, as multiple ways of comparison can be made.

We unintentionally missed to reference Figure S5A (Figure 2—figure supplement 1A) in revised manuscript in the text. We referred to Figure S5A in line 215. The significance shown in the Figure S5B (Figure 2—figure supplement 1B in revised manuscript) middle plot was calculated by comparing the proportion of each lineage cell of each genotype to that in GFP control. We described this point in the legend.

Figure 2D/E provided in vitro results for restoring CEBPa in rescuing the myeloid bias in the Pcgf1Δ/Δ HSPCs. Support this claim with in vivo results.

Thank you for your suggestion. We tried competitive BM transplantation assays using conditional *Pcgf1* KO, *Cebpa* KO, and compound mice to rescue dysregulated differentiation of *Pcgf1^Δ/Δ^* HSPCs. However, we found that the deletion efficiency of *Cebpa* was poor in vivo using tamoxifen, while it is quite efficient in vitro using 4-hydroxytamoxifen. The data we obtained was hard to interpret. We would like to address this issue in our future experiments using different approaches.

Line 242 refers to Figure S3D: there is no panel D in Figure S3.

Thank you for pointing this out. We corrected Figure S3D to Figure 1D.

Line 311 Figure 3D – this should have been 3G, if I understood correctly. If I did not understand correctly, then Figure 3G is never cited. This is so terribly confusing!

We apologize for the incorrect description. We corrected Figure 3D to Figure 3G.

Figure 3I – so poorly aligned/labelled.Line 320-322 "Moreover, ATAC-seq analyses also revealed trend toward enrichment of ATAC peaks containing CEBP motif in Pcgf1Δ/Δ HSPCs compared to the control, suggesting that chromatin regions containing CEBP motif tended to open in Pcgf1Δ/Δ HSPCs (Figure 3I)." The figure supporting this claim seems to have "p=0.3". Such a p-value is "ns" everywhere else. Why is it different here? As far as this reviewer can gather, the only significant difference in this figure is between control LSK and control GMP. Therefore, these results cannot and should not be cited as evidence to support "chromatin regions containing CEBP motif tended to open in HSPCs (Figure 3I)." Authors should remove any misleading claim based on this strenuous comparison.

We are sorry for misleading statements. We deleted Figure 3I and related statements.

Figure 4A. Can the difference in cluster GMPs be quantified?

We quantified cKit^+^FcγR^+^ cells (GMP clusters) using ImageJ software and presented the data in Figure 4A right panel.

Line 412-413 figure legend for Figure 4C: "Each symbol is derived from an individual culture". This is supposed to be from in vivo experiments? No statistical significance indicated?

Thank you for pointing this mistake out. We corrected “an individual culture” to “individual mice”. We have already shown statistical significance against day 0 for each genotype. For example, in the case of total BM, statistical significance was confirmed only on day 8 for both Control and *Pcgf1^Δ/Δ^*. We described these points in the legend.

Figure 5B,5G/H, states that HSC cultures produced GMPs. This statement is misleading and contributes to the over-interpretation of the results. To the best that this data can support, perhaps it is fair to say cells expressing immunophenotypic GMP markers are produced. The authors interpreted the difference in myeloid culture condition 1 and 2 being the different responses of HSC and GMP toward the loss of Pcgf1. However, it is equally possible that the different growth factors (type and concentrations) caused the differential response between the control and Pcgf1Δ/Δ cells. The authors need to culture HSC and GMP in both myeloid culture condition 1 and 2 to justify their claims.

Thank you for your suggestions. We changed the designation from GMP to immunophenotypic GMP. We also performed GMP culture in Figure 5A using myeloid culture condition 2, the same condition as HSCs in Figure 5B, and presented new data in Figure 5A.

Figure 5D. The images are missing the labels of genotypes.

These images are included simply as examples of nuclear β-Cat determination. All images are from control cells. We indicated this information in its legend.

Figure 5E legend says the samples are the GMPs from HSC cultures in 5B. Were these "GMPs" sorted? Please specify. Why are the day 16 control "GMPs" missing?

We sorted immunophenotypic GMPs (CD34^+^FcγR^+^c-Kit^+^Sca-1^-^Lineage^-^). We clarified this point in the legend. Unfortunately, control GMPs on day 16 were too few to analyze.

Figure 5F. What samples are being compared by RNA-seq? This reviewer is guessing the same samples shown in Figure 5E. If so, it begs the same question as to why are the day 16 control "GMPs" missing, and why are the day 0 Pcgf1Δ/Δ GMPs missing.

We are sorry for the lack of clarity, All samples were compared to the day 0 control GMPs. It is indicated in the upper left corner of the table. The day 16 control GMPs was too few to perform analysis. This is why we used day 0 *Pcgf1^Δ/Δ^* GMPs as a reference.